# A transposon expression burst accompanies the activation of Y-chromosome fertility genes during *Drosophila* spermatogenesis

Matthew A. Lawlor [1], Weihuan Cao[1] & Christopher E. Ellison [1✉]

Transposable elements (TEs) must replicate in germline cells to pass novel insertions to offspring. In *Drosophila melanogaster* ovaries, TEs can exploit specific developmental windows of opportunity to evade host silencing and increase their copy numbers. However, TE activity and host silencing in the distinct cell types of *Drosophila* testis are not well understood. Here, we reanalyze publicly available single-cell RNA-seq datasets to quantify TE expression in the distinct cell types of the *Drosophila* testis. We develop a method for identification of TE and host gene expression modules and find that a distinct population of early spermatocytes expresses a large number of TEs at much higher levels than other germline and somatic components of the testes. This burst of TE expression coincides with the activation of Y chromosome fertility factors and spermatocyte-specific transcriptional regulators, as well as downregulation of many components of the piRNA pathway. The TEs expressed by this cell population are specifically enriched on the Y chromosome and depleted on the X chromosome, relative to other active TEs. These data suggest that some TEs may achieve high insertional activity in males by exploiting a window of opportunity for mobilization created by the activation of spermatocyte-specific and Y chromosome-specific transcriptional programs.

[1] Department of Genetics, Human Genetics Institute of New Jersey, Rutgers University, Piscataway, NJ, USA. ✉email: chris.ellison@rutgers.edu

Transposable elements (TEs) are abundant in the genomes of plants and animals despite the presence of sophisticated host genome defense pathways. The genetic mechanisms responsible for the evolutionary success and persistence of TEs remain unclear. It is possible that the fitness benefit of complete TE suppression is not large enough to be evolutionarily favorable[1–3]. On the other hand, it is also possible that, like many viruses, TEs are engaged in an evolutionary arms race with their hosts, with TEs continuously evolving to escape silencing and the host genome continuously evolving to reestablish TE suppression[4]. Many host genes involved in TE defense are rapidly evolving, consistent with ongoing host-TE conflict[5–9], however relatively few strategies where TEs can escape or evade host silencing have been identified[10]. In the *Drosophila* ovary, there is evidence that some TEs propagate in permissive nurse cells and hijack the host's mRNA transport pathway to move to the developing oocyte, which is more recalcitrant to TE expression[11]. In another study, Dufourt et al. identified a small region of mitotically dividing germline cysts where the piRNA pathway effector protein Piwi is depleted and TE silencing is much weaker than in the surrounding cells. They termed this region the "piwiless pocket" and proposed that TEs may take advantage of this niche to replicate in the *Drosophila* germline[12,13].

TE replication and host silencing have been extensively studied in the *Drosophila* ovary, however surprisingly little is known about these same phenomena in the testes. Several previous observations suggest that there may be substantial differences between ovaries and testes with respect to both TE activity levels and host silencing pathways. For example, the expression patterns of multiple TE families are known to exhibit strong sex biases: The *I-element*, *P-element*, and *gypsy* TE families are all expressed at higher levels in the female germline[14–16], whereas the opposite is true for the *copia*, *micropia*, *1731*, and *412* TE families[17–20]. The piRNA pathway is active in both somatic and germline cells in the ovary and piRNAs bound by *Aub* and *Ago3* undergo robust ping-pong amplification in the ovarian germline. In the testes, TE-derived piRNAs are produced in germ cells, however the vast majority (~75%) arise from the *suppressor of stellate* [*Su(Ste)*] and *AT-chX* satellite repeats, rather than the canonical piRNA clusters that have been identified in ovaries[21,22]. Furthermore, many TE families show large differences in piRNA abundance between ovaries and testes[22] and TE-derived piRNAs only show a weak signature of ping-pong amplification in spermatocytes, likely due to low levels or absence of *Ago3*[21].

Here, we analyze TE expression at single-cell resolution to gain insight into the dynamics of TE activity in *Drosophila* testes. Our method for identification of TE and host gene co-expression modules shows that a subset of primary spermatocytes expresses a diverse group of TEs at high levels relative to other cell types. These TEs are co-expressed with Y-linked fertility factors and we find evidence that they are more active in males compared to females. These data suggest some TEs may exploit spermatocyte-specific transcriptional programs and Y-chromosome activation to remain active in the *Drosophila melanogaster* genome.

## Results

**Data processing and cell-type identification.** We reanalyzed 10× Genomics 3′ single-cell expression data from a recent study examining sex chromosome gene expression in larval testes from *D. melanogaster* strain w1118[23]. The *Drosophila* larval testes are elongated spheres encased in epithelial cells. The cell types of the testes are ordered spatially along a developmental gradient: The apical cap contains germline stem cells and somatic hub cells. The germline stem cells give rise to daughter cells that undergo mitosis with incomplete cytokinesis to create cysts containing

16 spermatogonial cells. Each cyst is encapsulated by somatic cyst cells and the spermatogonial cells further differentiate to form meiotic spermatocytes. In the larval testes, the primary spermatocytes exist in an extended meiotic prophase and do not undergo further differentiation.

We clustered cells based on their gene expression profiles to create cell clusters (Fig. 1a) and assigned each cluster to a known testis cell-type based on the expression of curated marker genes (Fig. 1b) (see Methods, Supplementary Table 1). Our filtering approach is more conservative than applied to these data in their initial study, yielding a final dataset with fewer cells than originally published (Supplementary Fig. 1A). There is a strong correspondence between our clusters and those previously identified from these data, with high pairwise correlations for every cluster previously reported, though minor differences are apparent (Supplementary Fig. 1B). Notably, we identify fewer distinct cyst cell clusters but more distinct spermatocyte clusters than reported in the original study.

Similar to the previously described analysis of these data[23], we identify distinct somatic and germline clusters (Fig. 1c). We identify cyst cells (clusters 7 and 8) which express *tj* and *wnt4* at high levels. Terminal epithelial cells (cluster 10) are defined largely by *Fas3* expression, and pigment cells (cluster 9) express *Sox100B* (Fig. 1b)[23].

The remaining cells comprise the germline components of these data. Cluster 1 contains early spermatogonia, marked by *vasa* and *spn-E* (Fig. 1b). Cluster 2 is most transcriptionally similar to the G spermatogonia cluster identified by Mahadevaraju et al. but mean normalized unique molecular identifier (UMI) counts for this cluster also correlate well with that study's E1 early spermatocyte cluster (Supplementary Fig. 1B). Furthermore, this cluster expresses spermatogonial markers such as *bam* as well as spermatocyte markers such as *aly*, which respectively are required for germline stem cell differentiation and initiation of a primary spermatocyte transcription program (Supplementary Fig. 1C). This observation suggests that our cluster 2 may represent spermatogonia just beginning the transition to meiotic prophase or very early spermatocytes. We therefore refer to cluster 2 cells as transitional spermatocytes.

The final four clusters (3, 4, 5, and 6) represent the remaining filtered cells (Fig. 1c) and express *aly* as well as *sa* and *can*, which are effectors of the primary spermatocyte expression program[24,25] (Fig. 1b). These clusters are transcriptionally similar to primary spermatocytes identified previously (Supplementary Fig. 1B). Mean expression in clusters 3 and 4 correlates well with the previously reported early primary spermatocytes while expression in clusters 5 and 6 correlates most highly with previously reported middle and late primary spermatocyte clusters (Supplementary Fig. 1B). Taken together, these observations suggest that the germline clusters may be ordered from earliest to latest differentiation state by the cluster numbers reported here. However, among the later putative spermatocyte clusters (4, 5, and 6) it is challenging to definitively identify the differentiation order.

**A spermatocyte subpopulation shows high expression of transposable elements**. In addition to the single-cell gene expression profiles, we also quantified TE expression in each cell (see Methods). We then visualized cell-type-specific expression patterns of all TEs with at least three UMIs detected across all individual cells in the *Drosophila* testis (Fig. 2a). Most striking are the cells from cluster 3 spermatocytes, where 28 TE families show expression in more than half of the cells in the cluster (Fig. 2b). In comparison, only two TE families are expressed by at least half of cluster 1 spermatogonial cells and only one TE family shows

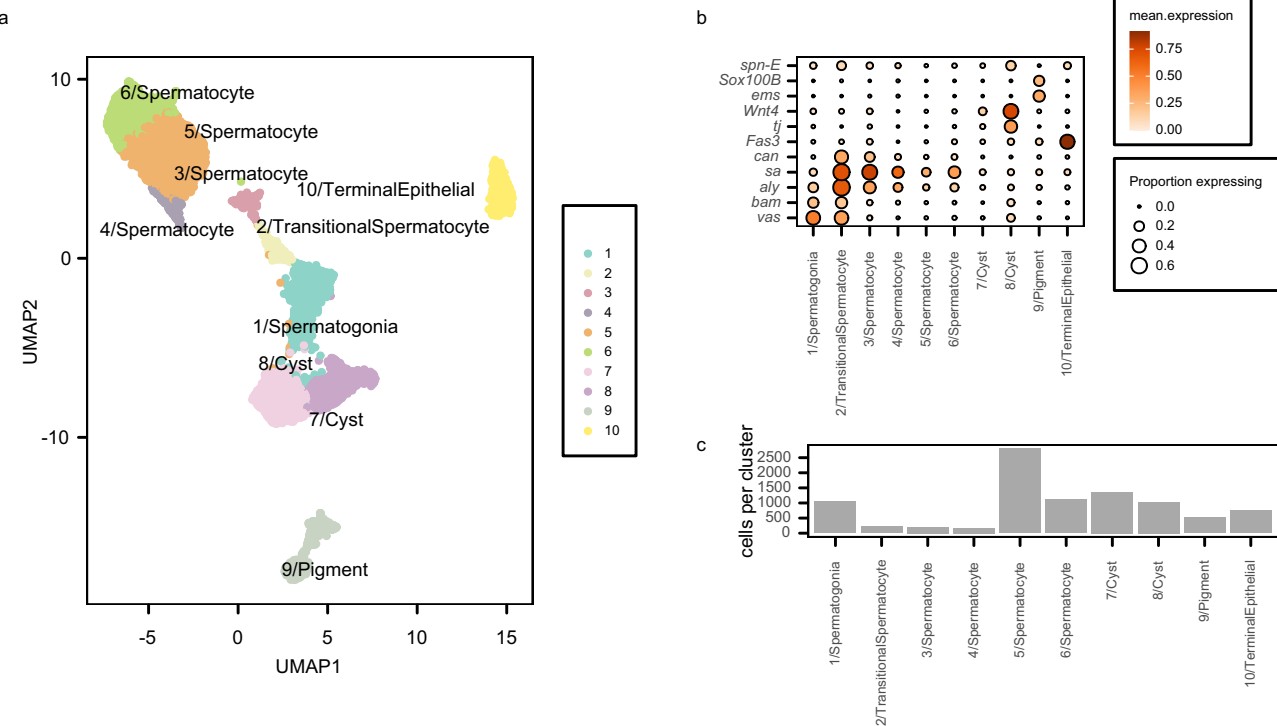

**Fig. 1 Identification of testis cell populations. a** Uniform Manifold Approximation and Projection (UMAP) groups transcriptionally similar cells in 2D space. Cells are colored by assigned cell type. **b** Dot plot shows expression of selected marker genes used for cell-type assignment. Color of each dot corresponds to mean normalized and log-transformed expression within cell clusters. Dot size corresponds to the proportion of cells in each cluster expressing the marker gene. **c** Cell counts within each cell-type cluster. Source data are provided as a Source Data File.

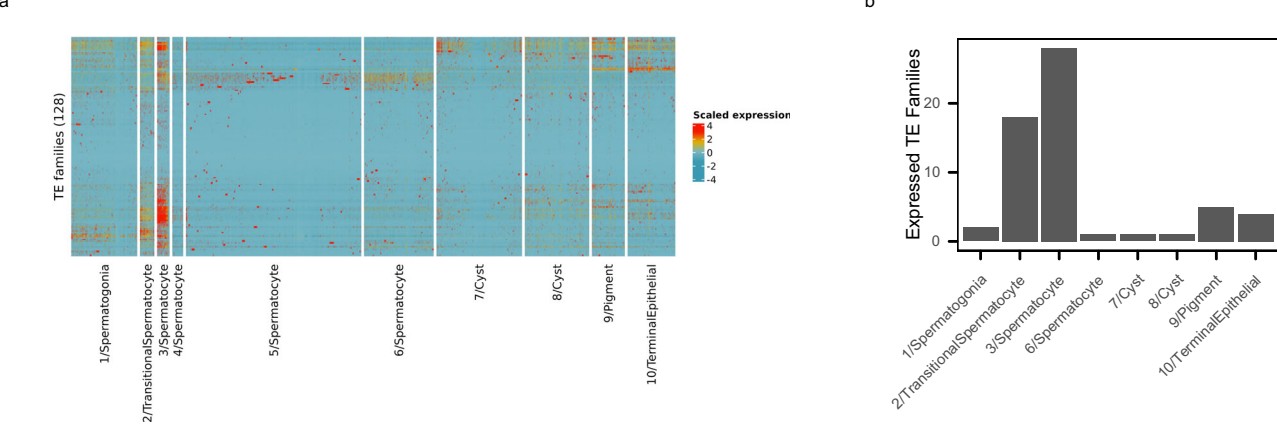

**Fig. 2 A spermatocyte cluster expresses transposons at high levels. a** Heatmap shows scaled expression levels of 128 transposable element (TE) families detected in this dataset across all cells. **b** The numbers of TE families with detectable expression in at least half of the cells in each cluster is shown. While most clusters express small groups of transposable elements or show sporadic expression of transposons in some member cells, cluster 3/Spermatocyte cells show high expression of many transposable elements. Cluster 2/Transitional Spermatocytes express the second highest number of TEs (18 families), 14 of which are shared with cluster 3. This pattern is consistent with a burst of TE expression that coincides with the developmental transition from spermatogonia to spermatocytes. Source data are provided as a Source Data File.

expression in at least half of each cyst cell cluster (Fig. 2b). Four and five TE families are expressed by at least half of Terminal Epithelial cells or Pigment cells respectively, of which 3 families overlap (Fig. 2b). Interestingly, the cluster 2 transitional spermatocytes have the next largest number of expressed TE families (18 families expressed by more than half of cells, Fig. 2b) and there is a high degree of overlap between the families expressed in cluster 2 and cluster 3 (14 TE families expressed in more than half of cells in both clusters), consistent with the transcriptional activation of these TEs families coinciding with the

developmental transition from spermatogonia to spermatocytes. However, cluster 3 spermatocytes have the most TE-derived UMIs per cell, for both depth-normalized and raw UMI counts (Supplementary Fig. 2A, B). Notably, cell clustering was performed using only expression measurements from highly variable host genes which suggests that cluster 3 is transcriptionally distinct from other spermatocytes independent of TE expression.

To verify that the detected TE expression pattern is not a technical artifact of 10X scRNA-seq, we analyzed larval testis poly-A selected bulk RNA-seq reads generated from the same

strain alongside the single-cell data[23]. TE expression levels from the bulk RNA-seq data are highly concordant with pseudobulk levels from the scRNA-seq data, both globally and with respect to TEs specifically (Supplementary Fig. 2C, D). We next assessed whether TE fragments nested in other cellular RNAs may be artificially increasing measurements of TE expression in the testes. If a TE fragment were present within a highly expressed host gene (in the UTR, for example), the RNA-seq reads from the fragment would get mapped to the TE consensus sequence, thus artificially inflating the expression level for that TE. We used two approaches to determine whether this phenomenon was affecting our estimates of TE expression: (1) we explicitly searched for chimeric RNA-seq reads, where part of the read aligned to a TE and another part aligned to a host gene, and (2) we examined RNA-seq read coverage along the TE consensus sequence. If TE fragments and/or host gene/TE fusions are a major source of RNA-seq reads, the full-length TE consensus should show non-uniform sequencing coverage with the region corresponding to the TE fragment showing much higher coverage than the rest of the TE. While a small number of families (2 out of 125 families analyzed) exhibit extreme coverage at localized portions of their consensus sequence, consistent with truncated copies and/or host gene-TE fusions, the vast majority of TE families expressed in testis show coverage throughout their consensus sequences and within-TE RNA-seq signal variability is comparable to single isoform host genes (Supplementary Fig. 2E, F). We additionally queried the chimeric reads identified by the STAR aligner[26] from w1118 larval testis poly-A selected bulk RNA-seq dataset (the same strain used for the scRNA-seq data). Only a small number of TEs show evidence of reproducible chimeric transcripts (Supplementary Fig. 2G, H).

**A TE-enriched gene module is expressed in spermatocytes.** To gain additional insight into the biological context involving the upregulation of TEs in cluster 3 spermatocytes, we used the single-cell expression profiles of both host genes and TEs to infer co-expression modules. Co-expression modules are groups of genes and/or TEs with correlated expression patterns. Members of the same expression module are frequently co-regulated and member genes often have related functions. Clustering-based algorithms are commonly used for the identification of modules, however clustering approaches usually examine co-expression across all samples, which is not ideal for single-cell expression data, where co-expression patterns may be limited to specific cell types. For this reason, we decided to infer gene and TE co-expression modules using an approach that can identify local co-expression signatures existing in only a subset of cells. Independent Component Analysis (ICA) has previously been shown to have this property and, in general, it performs favorably compared to other module detection methods[27].

We applied a consensus ICA approach[28] and selected parameters that resulted in modules representing a wide range of biological processes while also showing minimal overlap in terms of their gene content (see Methods). The modules resulting from this approach were reproducibly identified across replicate runs of consensus ICA (Supplementary Fig. 3A) and ranged in size from 10 to over 600 genes, with 72% (65 out of 90) of identified modules containing 200 or fewer genes (Supplementary Fig. 3B). 64% (58 out of 90) percent of identified modules were enriched at $p < 0.05$ for a Biological Process Gene Ontology (GO) term not enriched in any other module (Supplementary Fig. 3C).

Our method identified many co-expression modules with at least one TE included alongside host genes, but a single module (module 27) included over 70 transposons along with approximately 300 host genes (Fig. 3a, b, Supplementary Data 1, Supplementary Data 2). All major classes of TEs are represented in this module, including LTR and non-LTR retroelements and DNA transposons. The majority of these TEs (73 out of 75) are likely to be currently or recently active in *D. melanogaster* because they were identified as having polymorphic insertions in the TIDAL-FLY database[29].

Several TEs in this module have previously been shown to have male-biased expression: the LTR retrotransposons *1731*, *412*, and *copia* are expressed at high levels in the primary spermatocytes of *D. melanogaster*[18–20], while *micropia* transcripts have been shown to be associated with Y-chromosome lampbrush loops in the primary spermatocytes of *D. hydei*[17]. We visualized per-cell expression scores (see Methods) for module 27 on the UMAP projection and observed that it is expressed mainly by cells in cluster 3 and a small number of cluster 2 cells, in agreement with our visual inspection of TE expression across the dataset (Fig. 3c). These results suggest that a burst of TE expression occurs in a distinct subcluster of primary spermatocytes in the larval testes.

We identified the host gene *EAChm* as a marker gene that shows high specificity for module 27-expressing spermatocytes (Fig. 4a). EAChm is an enhancer of *chm* acetyltransferase activity that shows high expression in adult testis in modEncode poly-A selected bulk RNA-seq data[30,31]. Its role in spermatogenesis is currently unknown. To confirm co-expression of module 27 host genes and TEs in 3/Spermatocyte cells, we performed multiplexed RNA-FISH in whole-mount larval testes for *EAChm* and two module 27 TEs: *ACCORD2* and *QUASIMODO2* (Fig. 4b). We find that *EAChm*, *ACCORD2*, and *QUASIMODO2* show similar spatial patterns of expression, consistent with their membership in the same co-expression module (Fig. 4b, Supplementary Figs. 4, 5, 6). Furthermore, the transcripts of all three elements are confined to the central portion of the larval testis, in agreement with our assessment that module 27-expressing cells are primary spermatocytes.

We next sought to determine whether the same gene module is expressed in the testes of adult flies. To do so, we reanalyzed previously published single-cell RNA-seq from adult testes of a different *D. melanogaster* strain[32]. The TE expression profile from our 3/Spermatocyte cells that express module 27 is highly correlated with a putative spermatocyte cluster we identified in the Witt et al. data, suggesting that the members of module 27 are also co-expressed in the spermatocytes of adults as well as other strains of *D. melanogaster* (Spearman's $R = 0.49$, $P = 9.3e-6$) (Supplementary Fig. 7).

We next examined the genes in module 27 and found several primary spermatocyte-restricted genes that are required for sperm maturation (Supplementary Fig. 8A). Two testis-specific TBP associated factors (TAFs), *can* and *sa*, are members of module 27 and are differentially expressed (*T* test, two-sided adjusted $P < 0.05$) in cluster 3. Testis-specific Meiotic Arrest Complex (tMAC) components *aly* and *wuc*, which promote transcription of spermatocyte-specific genes by activating alternative promoters[33], are members of module 27, as well as *kmg*, which blocks promiscuous activation of genes by tMAC[34]. Each of these is also significantly differentially expressed in cluster 3 (*T* test, two-sided adjusted $P < 0.05$) (Supplementary Data 3). This supports our analysis suggesting cluster 3 is predominantly composed of primary spermatocytes.

GO enrichment analysis shows that module 27 is enriched for genes that function in axonemal assembly and cilium movement, including the Y-chromosome fertility factors *kl-2*, *kl-3*, and *kl-5*, which are expressed specifically in primary spermatocytes[35] (Supplementary Fig. 8B). Overall, module 27 is significantly enriched for genes from the Y chromosome: of the 11 Y-chromosome genes used for module detection, 6 are assigned to this module (Supplementary Fig. 8C, Fisher's Exact test, two-sided $P = 1.7e-05$).

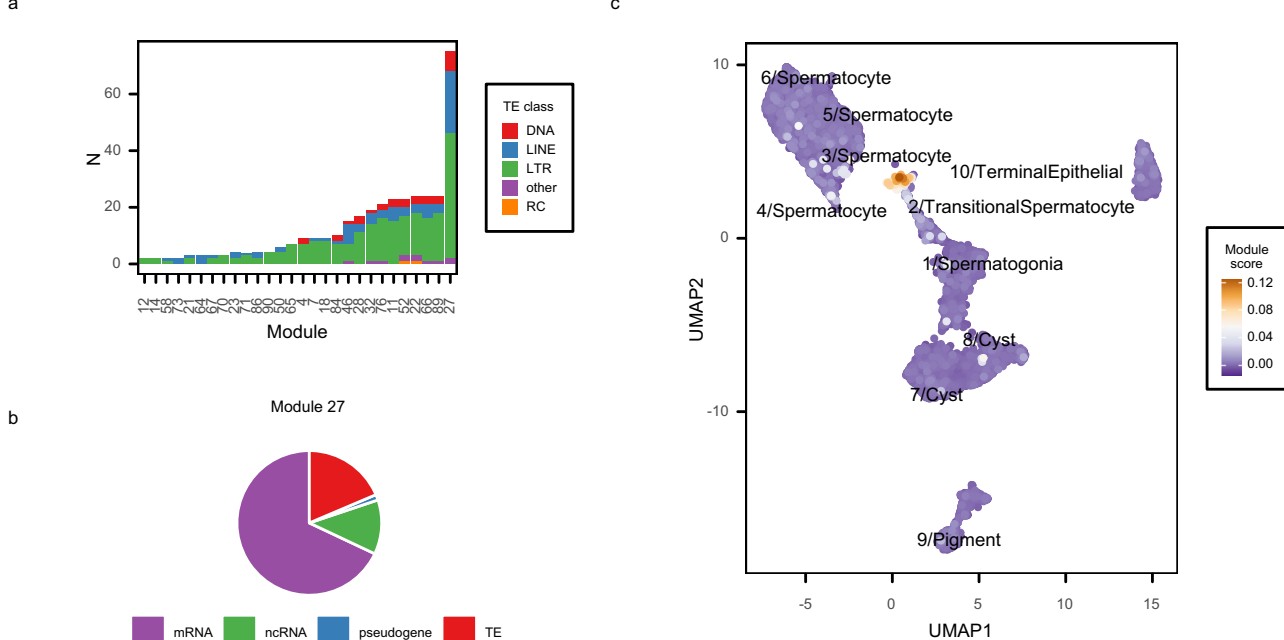

**Fig. 3 A TE-enriched expression module is transcribed primarily in cluster 3/Spermatocytes. a** Tallies of transposable element (TE) classes found in each module containing at least 1 TE. Module 27 contains almost fourfold more TEs than the next most TE-rich module and is predominantly composed of LTR retrotransposons (59%, 44 out of 75), LINE (29%, 22 out of 75), and DNA (9%, 7 out of 75) elements. **b** Module 27 contains over 300 features including protein-coding genes (68%), TEs (18.6%), or non-coding RNAs (12%). **c** UMAP embedding colored by module 27 expression score. Expression score is derived from the consensus Independent Component Analysis source matrix (see Methods). Source data are provided as a Source Data File.

In meiotic prophase, 16-cell primary spermatocytes undergo chromatin decondensation and greatly increase in size[36]. Y-chromosome lampbrush loops also form at this stage of development and the Y chromosome becomes enriched for the H3K9ac histone modification, which is associated with active transcription[37]. Consistent with this phenomenon, we also find that *tplus3a* and *tplus3b*, two genes required for expression of Y-chromosome fertility factors[38], are members of module 27 as well as *bol*, which binds the decondensed giant introns of several Y loop-forming genes[39]. All nine genes (6 Y-linked genes, *bol, tplus3a,* and *tplus3b*), are differentially expressed (*T* test, two-sided adjusted *P* < 0.05) in cluster 3 spermatocytes (Fig. 4c, Supplementary Fig. 8A, Supplementary Dataset 3). These results are consistent with the burst of TE activity that we observe in cluster 3/Spermatocyte cells coinciding with the activation of the Y-chromosome fertility genes.

**Y-chromosome activation and host defense downregulation coincide with module 27 TE expression.** Given that the module 27 TEs are co-expressed with Y-chromosome fertility genes, we hypothesized that their upregulation is due to activation of Y-linked copies of these TEs. To address this hypothesis, we first investigated whether these TEs do indeed have copies that are located on the Y chromosome.

We first used RepeatMasker to identify transposon insertions in a recently published *Drosophila melanogaster* Iso1 strain genome assembly with improved Y-chromosome content[40] compared with the current *D. melanogaster* Release 6 reference sequence. We found that 76% (57 out of 75) of module 27 TEs have at least one full-length copy located on a known Y-linked scaffold (Supplementary Fig. 8D) and a significantly larger percentage of module 27 TE insertions are found on the Y chromosome compared with other expressed TEs (Chi-square test one-sided *P* = 2.29e−292, Fig. 5a). We also estimated male-specific TE copy numbers by performing Illumina whole-genome sequencing (WGS) of males and females from strain w1118. We

found that module 27 TEs have significantly elevated copy numbers in males, compared to females, as expected if these TEs have insertions located on the Y chromosome (Wilcoxon Rank-Sum test, two-sided *P* = 0.0035, Fig. 5b).

Given that module 27 TEs are enriched on the Y chromosome, we next assessed whether their Y-linked copies are overexpressed in testes relative to their autosomal and X-linked copies. We used male and female WGS reads from w1118 to identify male-specific (i.e., Y-linked) single-nucleotide variants in module 27 TEs. We then compared the relative abundance of each male-specific variant in the w1118 ribosomal RNA-depleted bulk RNA-seq data from adult testes to its relative abundance in male WGS data (see Methods). A ratio larger than 1 indicates the presence of one or more Y-linked TE insertions that are expressed more highly than total-copy number alone would explain. For each expressed TE, we found the site of the male-specific allele most overexpressed relative to WGS depth. At these sites, male-specific module 27 TE alleles have significantly higher ratios of relative RNA to DNA coverage compared to male-specific alleles from TEs not present in module 27 (Wilcoxon Rank-Sum test, two-sided *P* = 1.7e−07, Fig. 5c). To confirm that this effect is specific to Y-linked module 27 TEs, we repeated our analysis using reference sites as well as autosomal (i.e., present in both males and females) variants. Contrary to the Y-linked TEs, these were expressed proportionately to WGS depth with no difference in expression proportion between module 27 TEs and other TEs (Wilcoxon Rank-Sum test, two-sided *P* = 0.24, Supplementary Fig. 8E).

Interestingly, the module 27 TEs are enriched for elements located within the *flamenco* piRNA cluster, which is involved in TE suppression in ovarian follicle cells[41] (Fisher's Exact Test, two-sided *P* = 0.03) (Supplementary Fig. 9A). One such TE is *gypsy*, which is both silenced by *flamenco* and is also a member of module 27. *Gypsy* has been previously reported to be enriched on the Y chromosome of several *D. melanogaster* strains, which Chalvet et al. proposed was because Y-linked copies of this TE

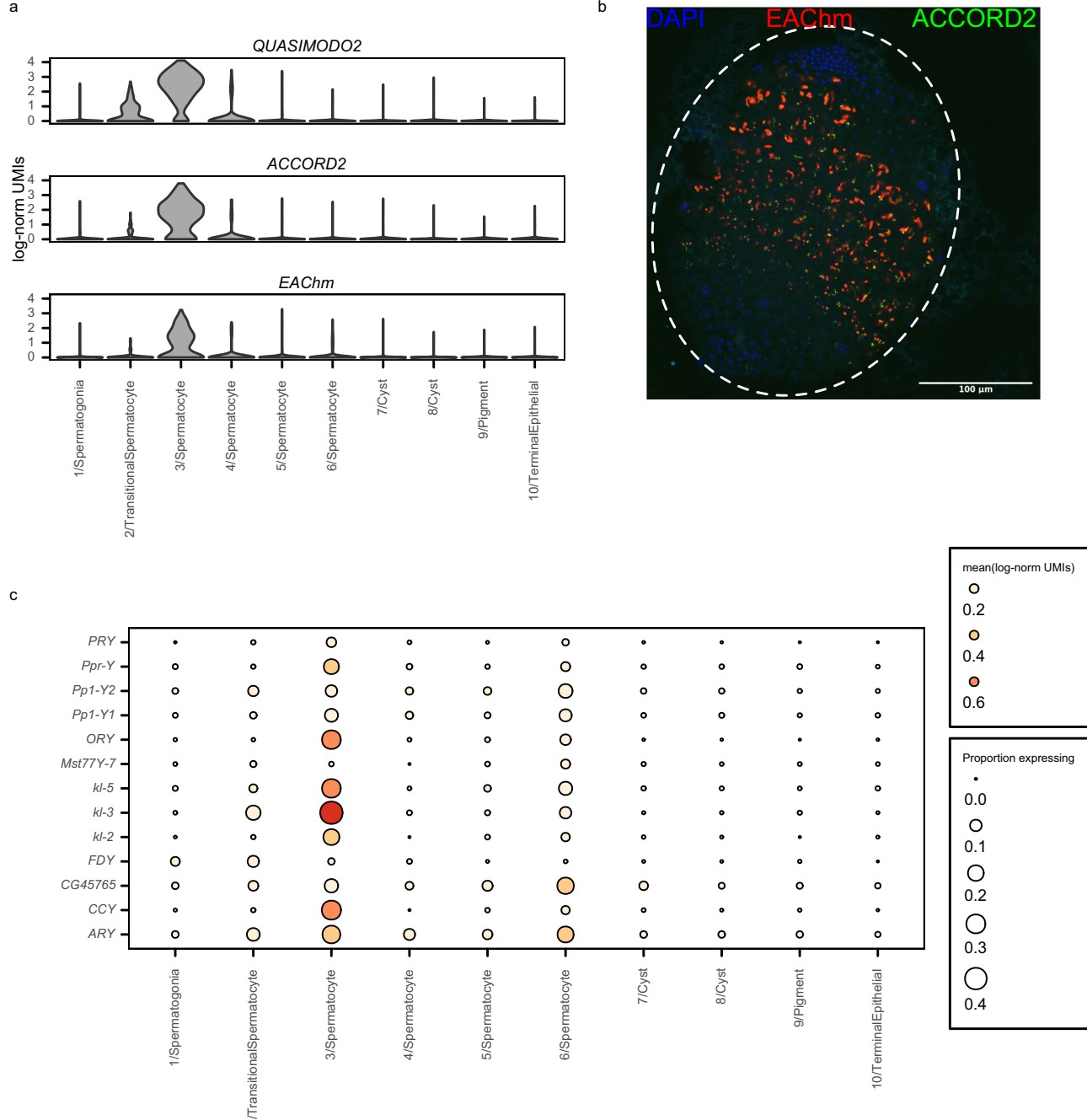

**Fig. 4 Module 27 expression co-occurs with Y-chromosome transcriptional activity. a** Violin plot shows that normalized expression of *EAChm*, *QUASIMODO2*, and *ACCORD2* is largely confined to cluster 3/Spermatocytes. **b** Multiplexed RNA fluorescence in situ hybridization (RNA-FISH) in whole-mount 3rd larval instar w1118 testis shows *ACCORD2* and *EAChm* expression is detected in the middle region of the testis, where primary spermatocytes are located. Red: *EAChm*; green: *ACCORD2*; blue: DAPI. Brightness and contrast were adjusted separately for each channel. Imaging was repeated in multiple testes with similar results. **c** Dot plot shows mean normalized expression of Y-linked genes in each cluster. Dot size corresponds to the proportion of cells in each cluster with detectable expression of each Y-linked gene. Y-linked genes, especially fertility factors *kl-3* and *kl-5*, are highly expressed by cluster 3/Spermatocytes. Source data are provided as a Source Data File.

were able to escape silencing by the ovary-dominant *flamenco* locus[42]. Consistent with such a strategy, module 27 is also enriched for TEs that are strongly silenced by the ovarian piRNA pathway in general (Fisher's Exact Test, two-sided $P = 7.6e-4$) (Supplementary Fig. 9B), raising the possibility that the expression of these TEs in the testes allows them to continue to mobilize in *D. melanogaster* despite being silenced in the ovary.

It is possible that our observed burst of TE expression is due to transcriptional readthrough of TE insertions present in the introns of Y-linked genes. To examine this possibility, we analyzed a heterochromatin-enriched *D. melanogaster* genome assembly[40] and found that module 27 TEs, while enriched on the Y chromosome overall, are significantly depleted from Y-gene introns when compared to other TEs with at least one detected Y insertion (Fisher's exact test, two-sided $P = 8.63e-10$, Supplementary Table 2). Our analysis of TE-gene fusion transcripts from poly-A selected bulk RNA-seq data did not detect any Y-linked TE-gene fusions, however it is possible that there are

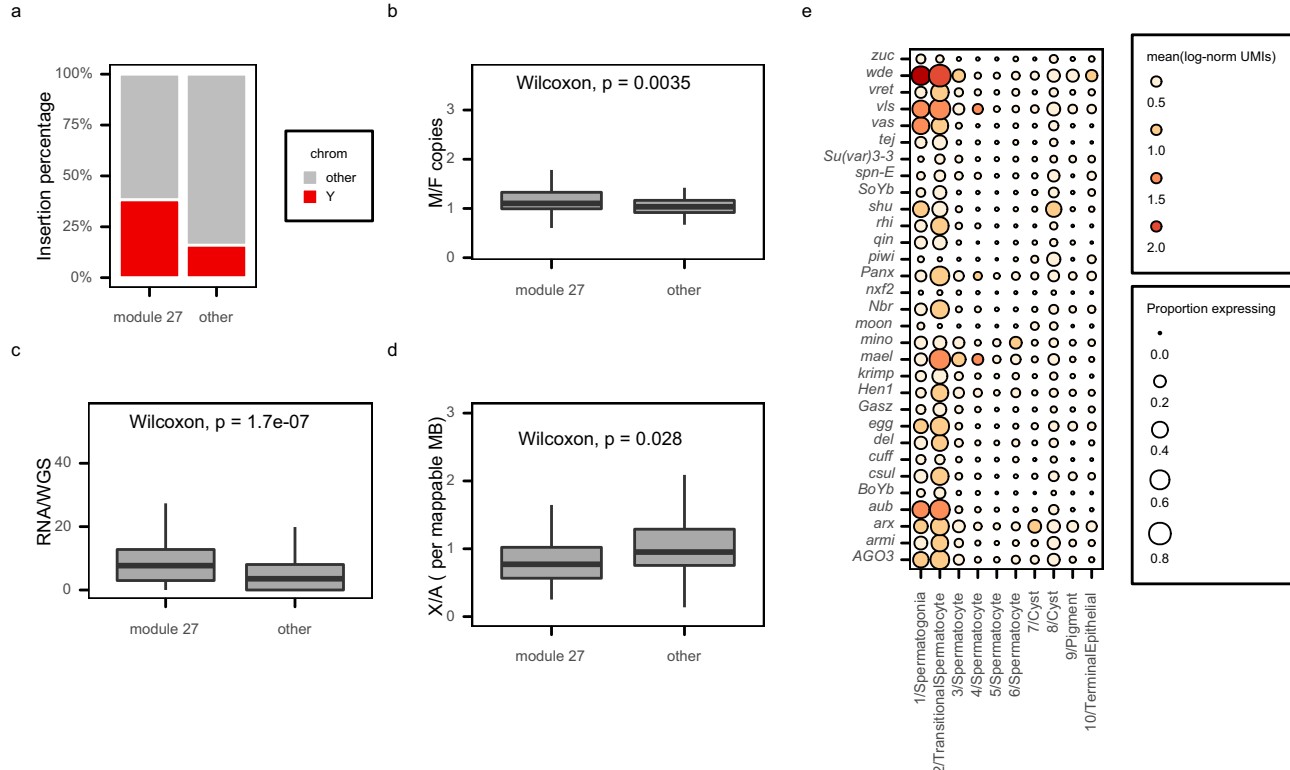

**Fig. 5 Module 27 Transposable elements are enriched on the Y chromosome. a** A higher proportion of module 27 transposable element (TE) insertions are found on the Y chromosome compared to TEs in other modules. Chi-square test, one-sided $P = 2.29e-292$. To better estimate Y-linked insertions despite the incomplete Y-assembly in the reference sequence, insertions were mapped from a heterochromatin-enriched assembly (see Methods). **b** Ratios of male and female copy numbers for individual TEs estimated from w1118 whole-genome sequencing (WGS) coverage. One male and one female sample were sequenced and used in ratio calculation. Module 27 TEs are present at higher copy number in the male genome. $N = 73$ module 27 TEs, 72 other module TEs. Wilcoxon rank-sum test, two-sided $P = 0.0035$. **c** Allele specific analysis of TE expression (see Methods) shows that Y-linked copies of module 27 TEs are overexpressed relative to their DNA copy number and this overexpression is significantly larger than that of TEs in other modules. $N = 58$ module 27 TEs, 32 other module TEs. Wilcoxon rank-sum test, two-sided $P = 1.7e-07$. **d** Boxplot showing the ratios of X-linked versus autosomal polymorphic insertions per mappable megabase for each TE in the TIDAL-fly database with a corresponding TE in this analysis. Module 27 TEs are depleted from the X chromosome compared to other TEs with polymorphic insertions. $N = 61$ module 27 TEs, 38 other module TEs. Wilcoxon rank-sum test, $P = 0.028$. For all boxplots, midline represents median, box represents interquartile range (IQR), and whiskers extend >1.5 IQR from the upper or lower quartile. **e** Dot plot shows expression of selected piRNA pathway genes. Color of each dot corresponds to mean normalized and log-transformed expression within cell clusters. Dot size corresponds to the proportion of cells in each cluster expressing the marker. Source data are provided as a Source Data File.

additional fusion transcripts which are not polyadenylated. To more exhaustively search for possible Y-linked TE-gene fusion transcripts, we also generated RNA-seq libraries from ribosomal RNA-depleted total RNA. We searched bulk total RNA-seq data from w1118 adult testes for TE-gene fusion transcripts and detected fusions for 5 TEs from module 27 (6.6% of all TEs in the module) of which only three are fused to a Y-linked gene that is itself a member of module 27. For these three TE-gene fusion breakpoints, supporting reads comprised fewer than 8% of spanning reads (Supplementary Fig. 10). Together these data suggest that the module 27 TE transcripts we detect in the scRNA-seq data are primarily generated via mechanisms other than readthrough of Y-linked genes.

We next investigated whether the module 27 TEs show increased insertional activity in males. Polymorphic TE insertions reflect recent TE insertions that are still segregating within a population. If the module 27 TEs replicate more often in males compared to females, recent polymorphic insertions of these TEs should be depleted from the X chromosome because this chromosome is hemizygous in males and is therefore a smaller mutational target. We used the TIDAL-FLY database of polymorphic TE insertions[29] for the *Drosophila* Genetic

Reference Panel (DGRP) to compare insertion frequencies of module 27 TEs versus other active TEs. We found that other TEs exhibit similar X and autosomal insertion rates across the DGRP lines whereas module 27 TEs exhibit a significantly reduced frequency of X-linked insertions relative to autosomal insertions, consistent with male-biased activity (Wilcoxon Rank-Sum test, two-sided $P = 0.028$, Fig. 5d).

It is also possible that Meiotic Sex Chromosome Inactivation, where the X chromosome is transcriptionally downregulated in primary spermatocytes[23], could further impede the ability of TEs to insert on the X chromosome during spermatogenesis. To assess this possibility, we compared frequencies of polymorphic TE insertions on the 4th chromosome to those on the autosomes. The 4th chromosome is the ancestral X chromosome in Dipterans and has also been shown to undergo transcriptional downregulation in *D. melanogaster* spermatocytes, along with the X chromosome[23]. However, in contrast to the X chromosome, we found that the module 27 TEs do not show a reduction in polymorphic insertions on the 4th chromosome (Supplementary Fig. 11). These results suggest that the reduction in X-linked insertions for the module 27 TEs is more likely to be due to the monosomy of the X, rather than X chromosome inactivation,

however, either mechanism is consistent with our conclusion that the module 27 TEs show male-biased activity.

*Ago3*, a piRNA pathway gene involved in the ping-pong piRNA amplification cycle, is present in germline stem cells and spermatogonia but undetectable in spermatocytes[21]. To determine whether there is a general trend of downregulation of piRNA pathway genes in spermatocytes compared to spermatogonia, we quantified expression of 31 piRNA pathway genes described in Czech et al.[43]. We found a clear trend showing a striking downregulation of most piRNA pathway genes during the developmental transition from spermatogonia to spermatocytes (Fig. 5e). Together, our results suggest that a burst of TE expression in *Drosophila* testes coincides with the activation of Y-chromosome fertility genes and the downregulation of piRNA pathway genes.

## Discussion

In *Drosophila* ovaries, constrained developmental processes such as the nurse cell to oocyte mRNA transport pathway create a window of opportunity that TEs have evolved to exploit in order to increase their own copy numbers[11]. Our results suggest a similar phenomenon has occurred in the testes, albeit via a different window of opportunity. A major source of TE activity in the testes is related to the presence of the Y chromosome itself. This chromosome acts as a safe harbor for TE insertions: The lack of recombination on the Y chromosome prevents efficient purging of Y-linked TEs from the population, allowing their accumulation along with other repetitive elements such as satellite DNA[44]. However, the Y chromosome usually exists as tightly packaged, transcriptionally silent, heterochromatin. How can the presence of this inert chromosome lead to TE activation? Interestingly, there is evidence that the Y chromosome can act as a "sink" for heterochromatin: its presence may cause a genome-wide reallocation of repressive histone modifications, which can lead to TE de-repression[45–48]. On the other hand, Wei et al. have recently described a phenomenon that they term "Y toxicity" based on the upregulation of TEs present on the neo-Y chromosome of *Drosophila miranda* during embryogenesis[49]. Their results suggest that transcription of the relatively large number of genes on the young neo-Y chromosome prevents complete silencing of this chromosome and therefore provides an opportunity for transcriptional activation of neo-Y-linked TEs.

Our results suggest that the Y toxicity phenomenon applies to older Y chromosomes as well. The ancient *Drosophila melanogaster* Y chromosome carries many fewer genes compared to the *D. miranda* neo-Y chromosome, however, at least six genes on the *D. melanogaster* Y chromosome are essential for male fertility[50–53]. These genes are known as fertility factors and they are only expressed during spermatogenesis[54]. The three annotated fertility factors, *kl-2, kl-3,* and *kl-5,* each span as much as 4 Mb due to their extraordinarily large introns and become transcriptionally activated in primary spermatocytes, which coincides with a general decondensation and acetylation of the Y chromosome[55]. The transcription of three of these genes, *kl-5, kl-3,* and *ks-1,* is associated with the formation of large Y-chromosome lampbrush loops[56]. The burst of TE expression that we describe here co-occurs with the activation of *kl-2, kl-3* and *kl-5,* as well as six other Y-linked genes: *ORY, ARY, Ppr-Y, Pp1-Y1, CG45765,* and *CCY.* Based on these results, we propose that the module 27 TEs have evolved to exploit a window of opportunity that occurs during the decondensation of the normally tightly packaged Y-linked chromatin, which is necessary for transcription of fertility factor genes.

Notably, not all TE families with intact Y-linked insertions are members of module 27, suggesting that additional features beyond Y-linkage, such as specific regulatory elements, are required for TEs to exploit this opportunity. Four module 27 TEs, *1731, 412, copia,* and *micropia* have previously been shown to be highly expressed in primary spermatocytes in *Drosophila* and *micropia* transcripts are physically associated with Y-chromosome lampbrush loops in *D. hydei*[17–20]. Interestingly, another 20 module 27 TEs, including *gypsy,* have insertions located within the *flamenco* piRNA cluster. Chalvet et al. identified multiple strains of *D. melanogaster* where active *gypsy* elements are confined to the Y chromosome[42]. They proposed that Y-linked *gypsy* insertions are able to evade silencing by the ovary-dominant *flamenco* locus, which may explain the enrichment of *flamenco*-regulated TEs among members of module 27. Indeed, more recent research has found that *flamenco*-derived piRNAs are almost an order of magnitude more abundant in ovaries compared to testes[22]. Altogether, 80% (60 out of 75) of the TEs in module 27 have at least one full-length copy within the assembled portion of the Y chromosome and/or are *flamenco*-regulated. This is likely an underestimate given estimates that ~60% of the Y chromosome remains unassembled[40].

*Flamenco* is not unique in terms of its female-biased piRNA production—the majority of known piRNA clusters produce more abundant piRNA in ovaries compared to testes[22]. Spermatocytes also lack a robust ping-pong amplification loop and the bulk of spermatocyte piRNAs come not from TEs, but rather two satellite repeats: *su(Ste)* and *AT-chX*[57]. Furthermore, piRNA factors such as Piwi and Ago3, while abundant in germline stem cells and spermatogonia, are missing or present at low levels in spermatocytes[21,57]. Our analysis of scRNA-seq data confirm these findings (Fig. 5e).

The downregulation of the piRNA pathway in spermatocytes is surprising. One would expect the piRNA pathway to be upregulated in these cells to safeguard against TE insertions resulting from de-repression of Y-linked TEs. So, why do spermatocytes show a weakened piRNA response at a developmental timepoint when the TE-rich Y chromosome is de-repressed? One possibility is related to intragenomic conflict. Sex chromosomes are hotspots for genomic conflict[49] and small RNA pathways may play an outsize role in defending against meiotic drivers in the male germline[58,59]. There is evidence that *stellate* and *su(Ste)* represent a cryptic meiotic drive system where X-linked *stellate* genes disrupt spermatogenesis and cause sex-ratio distortion in the absence of the Y-linked *su(Ste)* piRNA cluster[60,61]. The function of *AT-chX* is less clear. Although this locus was originally proposed to play a role in the developmental silencing of *vasa* during spermatogenesis[62], recent results instead suggest of role for *AT-chX* in hybrid incompatibility[63]. Neither locus is present in the genomes of close relatives of *D. melanogaster,* suggesting that they are dispensable for spermatogenesis[63,64]. The fact that both *su(Ste)* and *AT-chX* rapidly evolved to be essential for fertility in *D. melanogaster* is consistent with a role in mediating genetic conflict. This is especially clear for *su(Ste)* where the *stellate* protein is completely absent from wild-type flies[64]. If the *su(Ste)* and *AT-chX* piRNAs evolved to supress segregation distorters or other forms of selfish elements, it would suggest that there has been a tradeoff in the piRNA system in *D. melanogaster* spermatocytes, where increased abundance of *su(Ste)* and *AT-chX* piRNAs comes at a cost of impaired TE silencing.

Like *melanogaster,* *D. simulans* testes also show a weakening of piRNA-mediated TE silencing. *D. simulans* testes have been shown to have many fewer TE-derived piRNAs and higher TE expression compared to ovaries[65]. However, *D. simulans* lacks both the *su(Ste)* and *AT-chX* satellites. Thus, either the tradeoff between satellite- and TE-derived piRNAs is unique to *D. melanogaster* or a different tradeoff is present in *D. simulans,* such as between piRNA production versus other small RNAs. For example, siRNAs generated by hairpin RNAs have recently been

shown to play an important role in resolving intragenomic conflict in the *D. simulans* male germline[59]. Alternatively, there may be no tradeoff at all and the downregulation of the piRNA pathway during spermatogenesis may serve some other, as yet undetermined, developmental function.

In summary, this work provides detailed insight into the expression dynamics of TEs in the male germline of *Drosophila*. Our method for identifying gene/TE co-expression modules allowed us to identify a TE-rich expression module that is active in early spermatocytes. These results suggest that spermatocytes represent a permissive niche for TE expression due to the downregulation of piRNA pathway components and the transcriptional activation of the TE-rich Y chromosome in these cells. Additional work to improve the Y-chromosome assembly and uncover the mechanisms underlying the transcriptional activation of the Y chromosome will be critical for understanding the burst of TE expression that we have identified in spermatocytes. Furthermore, future work investigating the peculiarities of TE silencing in the testes will help shed light upon the tradeoffs and constraints imposed by the various roles of piRNAs in the male germline, including host gene regulation, TE silencing, and the resolution of intragenomic conflicts.

## Methods

**Repeat masking and custom reference sequence generation**. All repeat masking was performed with RepeatMasker[66] with the following options: "-e ncbi -s -no_is -nolow." We used RepBase *D. melanogaster* consensus TE sequences (version 20170127)[67] as a custom library.

All Illumina read alignments were performed using a custom reference sequence made by appending the consensus TE sequences to the TE-masked *D. melanogaster* r6.22 genome assembly.

**scRNA-seq processing**. Single-cell RNA-seq data was downloaded from PRJNA548742 and PRJNA518743 with wget v1.14[68]. We used 10X Genomics cellranger software to align and quantify the data[69]. We generated a cellranger index from the previously described custom reference sequence using cellranger's "mkref" command with default parameters. We aligned scRNA-seq reads using cellranger's "count" command with default parameters. We used cellranger's filtered count matrices for further analysis.

We first summed counts assigned to the LTR and internal sequences of class I LTR retrotransposons. For each scRNA-seq replicate, we next applied scrublet[70] v0.2.1 to these unnormalized count matrices to identify and filter putative heterotypic doublets. We used scanpy[71] v1.6.0 to retain genes detected in at least 3 cells and then cells with at least 250 and fewer than 5000 detected genes. We removed cells with more than 5% of remaining UMIs assigned to mitochondrion-encoded genes. We normalized UMI counts to 10000 per cell and applied log transformation with a pseudo-count of 1. We generated pseudobulk expression estimates by summing post-filtering counts across all cells and assessed the concordance of these normalized values with expression in four larval testis poly-A selected RNA-seq replicates[23] (Supplementary Fig. 2C, D).

We identified highly variable genes using scanpy's "highly_variable_genes" method with default parameters. We next scaled counts using scanpy's "scale" method and applied scanpy's "regress_out" method to remove count variance associated with cell cycle and mitochondrial UMI counts.

For each replicate, we used scanpy to perform principal component analysis on highly variable host genes and calculate nearest neighbor graphs using 15 principal components and 25 neighbors. We called cell clusters using the Leiden algorithm[72] via scanpy with a resolution parameter of 0.35. We combined all three larval scRNA-seq replicates using scanpy's "ingest" method and confirmed that each batch contributed cells to each cluster (Supplementary Fig. 12).

Automated cell-type assignment was performed using Garnett[73] v0.2.17 and a set of curated marker genes (Supplementary Table 1).

**Consensus ICA for module detection**. We chose ICA to identify gene expression programs because it performs highly with respect to recovering known functional gene modules and because it is easily adaptable to finding partially overlapping modules[27]. Some factorization approaches, such as ICA and non-negative matrix factorization (NMF) suffer from stochastically varying solutions. Kotliar et al. have previously introduced an elegant approach, termed consensus NMF (cNMF), to stabilize NMF solutions for scRNA-seq GEP detection[28]. This approach clusters the results of many iterations of NMF to buffer the influence of outlier solutions yielded by single runs of the algorithm. However, when we implemented this approach, we found that it yielded large GEPs utilized by broad cell types. We

therefore chose to use ICA factorization because this approach was able to group genes into smaller GEPs expressed specifically by smaller cell populations.

We applied a consensus approach similar to Kotliar et al. to ICA to address the issue of ICA solution randomness. We performed ICA factorization using expression values for highly variable genes identified by scanpy as well as the 128 TE families that passed initial filtering. Therefore, resulting modules may contain exclusively TEs or genes or a mix of both types of feature. We standardized the normalized, log-transformed expression matrix containing the highly variable genes identified by scanpy and TEs to have zero mean and unit variance. Standardized scores were clipped to a maximum absolute value of 10. Then, for any desired number of components *(k)*, our approach uses FastICA via sklearn[74] to decompose the standardized expression matrix 100 times, then concatenates the resulting gene *x* module matrices, partitions all modules into *k* clusters using k-means clustering, and averages the per-cell scores within each partition to yield a consensus cell *x* module matrix. The values in this matrix represent the expression of each module in each cell and we refer to these values as *module expression scores*. For the same partitions, we averaged per-gene scores from cell by module matrices to generate a consensus cell by module matrix. These values represent the strength of the evidence for co-expression of a given gene/TE with the other features in the module. We refer to these values as *module membership scores*.

We assigned genes to each program by applying fdrtool[75] to the vector of gene weights for each module. Genes with FDR q-values less than a desired cutoff *(q)* for each module were considered members of the module.

To obtain the two major parameters *(k* and *q)* used for this approach in an unbiased way, we applied a grid search approach to choose the two most important parameters of our pipeline (see *Module Detection Parameters* below).

To assess the reproducibility of the cICA module detection approach, we used the module membership scores for all genes present within a module. We then calculated Spearman's correlation coefficients between all pairwise combinations of modules from the three cICA replicate runs used to find parameters k and q (see "Module Detection Parameters" below). We found that all main result modules were significantly positively correlated (Spearman's rho > 0, $P < 0.05$) with a module in all grid search replicate runs. (Supplementary Fig. 3A).

The redundancy of modules detected for the main results was assessed by hierarchical clustering of gene-wise module membership scores or cell-wise expression scores derived from cICA (Supplementary Fig. 13A, B).

**Module detection parameters**. Use of ICA or other matrix decomposition approaches for gene program detection requires a priori assumptions about the optimal number of components (*k*) to request from the decomposition algorithm. Additionally, generation of discrete gene lists for each gene program requires application of arbitrary score cutoffs to determine program membership for each gene.

To reduce bias and use of arbitrary cutoffs, we used a grid search approach to choose *k* and the q-value cutoff for membership. Briefly, we ran consensus ICA (described above) in triplicate for combinations of q-value cutoffs between 0.005 and 0.1 and *k* between 10 and 120. We assessed the biological interpretability of the candidate solutions by enrichment for Gene Ontology Biological Process (GO:BP) terms. We found that optimizing only for maximum percentage of GO:BP-enriched GEPs yielded mostly large GEPs associated with very general biological processes. Under the assumption that a maximally interpretable set of GEPs should capture a wide range of biological processes and should favor discovery of minimally redundant GEPs, we then calculated two scores: one based on the breadth of GO:BP enrichments in a given ICA solution and the other on the unique assignments of GO terms to GEPs. We then rescaled these scores to a maximum of 1 and calculated a joint score by multiplying them together. For our final set of gene programs, we ran consensus ICA a final time with the *k* and *q*-value that maximized the average joint score across all three test replicates.

**Poly-A selected bulk RNA-seq from w1118 larval testes**. We trimmed poly-A selected RNA-seq (PRJNA475132) with fastp[76] v0.20.0 and aligned to the custom reference using STAR[26] v2.7.3 with chimeric junction detection turned on and "--chimScoreJunctionNonGTAG 0". Other non-default parameters used are available via the linked github repository. We identified evidence of chimeric TE transcripts from the "Chimeric.out.junction" output file. We calculated normalized coverage for each strand using deeptools[77] v3.3.1 "bamCoverage" command with "--smoothLength 150".

**Whole-genome sequencing library preparation**. 20 0- to 3-day-old w1118 males or females were collected on dry ice and then homogenized using an electric pestle. Qia-Amp DNA Micro kit was used according to instructions. DNA was diluted to 40 ng/μl in 55 μl of Elution Buffer and sheared in a Covaris sonicator with settings as follows: 10% duty cycle, 2.0 intensity, 200 cycles per burst, 1 cycle, and 45 s process time.

WGS library generation protocol was adapted from the Marshall-Lab DamID-seq protocol[78] available at marshall-lab.org. Briefly, sheared DNA was purified with homemade purification beads. End repair was performed with T4 DNA Ligase (NEB M0202S), T4 DNA Polymerase (NEB M0203S), PolI Klenow fragment (NEB M0210S), and T4 Polynucleotide kinase (NEB M0201S). Adenylation was

performed with 3′−5′ Klenow Fragment (NEB M0212L). Adaptors were ligated with NEB Quick Ligase for 10 min at 30 °C before two rounds of cleanup with homemade beads. NEBNext UltraII Q5 kit (NEB M0544) was used for PCR enrichment. A final round of cleanup with homemade beads was performed before quantification and sequencing.

**Whole-genome sequencing data processing**. We trimmed reads using cutadapt[79] v3.2.0 with options "-q 20 -m 35." We aligned trimmed reads with bwa-mem2[80] v2.0, removed duplicate reads with picard[81] v2.22.1 with option "VALI-DATION_STRINGENCY = LENIENT", and filtered out multimappers with samtools[82] v1.10.

To estimate TE copy numbers, we used mosdepth[83] v0.3.1 to calculate genome-wide read coverage in 100 bp bins for the autosomes and TE consensus sequences. We then calculated the median coverage value for each autosome (2L, 2R, 3L, 3R) and each TE consensus sequence. We averaged the median coverage across autosomes to produce a single coverage value for autosomes. We then used the ratio of TE coverage to autosome coverage to generate estimated copy numbers for all TEs. These estimated copy numbers were then used to compare male and female copy numbers.

We identified male-specific polymorphic sites with Rsamtools[84] by finding mismatches with a base quality of at least 10 and at least 15 supporting male reads but lacking supporting female reads.

**Ribosomal RNA-depleted bulk total RNA-seq from w1118 adult testes**. We used ~100 pairs of testes from 3 to 5-day-old mated w1118 males. The testes were dissected in 1X PBS and transferred into 200 µL RNAlater Solution. Tissue was pelleted by centrifuging at 5000 g for 1 min at 4 °C. Supernatant was removed and 300 µL 1× DNA/RNA Shield was added before homogenization with an electric pestle. Homogenized tissue was digested with Proteinase K at 55 °C for at least 30 min. RNA was purified with the Zymo Quick-RNA Plus Kit (R1057).

Using up to 5 µg total RNA, ribosomal RNAs were removed using iTools rRNA depletion Kit from Galen Laboratory Supplies (dp-P020-000007) and Thermo Fisher MyOne Streptavidin C1 Dynabeads (#65001). RNA Clean and Concentrator-5 kit from Zymo Research (R1015) was used to purify rRNA-depleted RNA. Starting with 1 ng-100 ng purified rRNA-depleted RNA, Illumina libraries were generated using NEBNext Ultra II Directional RNA Library Prep Kit for Illumina (E7760).

For quantifying Y-linked expression for TEs, raw reads were trimmed with fastp[76] v0.20.0. We used STAR[26] v2.7.5 to align total RNA-seq reads to a bait reference composed of Flybase release 6.22 tRNA sequences and miscRNA sequences. We then aligned unmapped reads to our custom reference and provided STAR with a VCF file containing male-specific variants.

For Y-gene fusion detection, raw reads were trimmed with fastp v0.20.0. We used STAR[26] v2.7.9 to align total RNA-seq reads to our custom genome sequence with parameters recommended by Arriba documentation. Arriba[85] v2.1.0 was run with blacklist, uninteresting_contigs, read_through, intronic, long_gap, and intragenic_exonic filters disabled and "-U 32766". Consensus TE sequence names were passed as viral contigs via the "-v" flag.

**DGRP polymorphic TE insertions**. Using the TIDAL-Fly polymorphic TE insertion database[29], we found the number of unique polymorphic insertions on the X chromosome and on autosomes, excluding chromosome 4, across the *Drosophila* Genome Reference Panel for all TEs in our custom reference. For all TEs with at least 1 X-linked and 1 autosomal insertion among all DGRP lines, we calculated the ratio of X-linked or chromosome 4-linked insertions per megabase to autosomal insertions per megabase. To correct for possible differences in read mappability between chromosomes, we used the mappable lengths of each chromosome as reported by TIDAL-FLY[29]. In cases where there were differences in the TE family names between RepBase and TIDAL-FLY, we used NCBI BLAST[86–88] Web v2.11.0 to confirm that the consensus sequences were identical or nearly identical and then treated the family names as synonyms.

**Analysis of TE enrichment on Y chromosome and in Y-gene introns**. We used RepeatMasker to identify insertions in a heterochromatin-enriched assembly[40] as described above for the main reference sequence. For overall Y enrichment, we did not consider the terminal repeat portions of LTR retrotransposons, as we wanted to test the enrichment of potentially active copies. For Y-gene intron enrichment, we considered only Y-linked insertions but we did not remove LTRs, as solo LTRs could potentially contribute to detected transcription.

**RNA-FISH**. Custom Stellaris FISH probes recognizing *EAChm* labeled with Quasar670 and against *ACCORD2* and *QUASIMODO2* labeled with CAL Fluor Red 610 were designed using Stellaris' probe design tool available at www.biosearchtech.com/stellarisdesigner. Default parameters were used for *EAChm* probes. Probes against *ACCORD2* and *QUASIMODO2* were designed with masking parameter 2. To ensure specificity of the resulting probes, we used BLAST[89] to align to consensus TE sequences used for masking and custom reference generation to ensure that all probes show complementarity to their intended target only. *ACCORD2* and *QUASIMODO2* probes were also blasted against *Drosophila melanogaster* REFSEQ sequences and any individual probes

with more than 16 nucleotide matches to another sequence were removed from the final probe set. Probe sequences are available in Supplementary Data 4.

Strain w1118 flies maintained at room temperature were mated for 4 h and offspring were grown at 25 °C until reaching the third instar. We dissected L3 males in sterile 1X PBS and fixed testis in 3.7% formaldehyde solution at room temperature for 45 min. Testis were washed twice with 1X PBS and submerged in 70% ethanol at 4 °C overnight. Hybridizations were carried out according to instructions available on the manufacturer website.

Image slices were captured on a Carl Zeiss LSM880 AxioObserver with a C-Apochromat 40×/1.2 W Korr FCS M27 water immersion objective. 2D deconvolution was performed using ZEN Black software. Further contrast adjustments and image overlays were performed with Fiji[90].

**Identification of *flamenco*-regulated TEs**. RepeatMasker results (see above) for dm6.22 were parsed with bedops[91] "rmsk2bed" and TEs with copies within the range X:21631891-21790731 were identified as *flamenco*-regulated.

**Identification of ovary piRNA pathway-regulated TEs**. RNA-seq reads for piRNA pathway component knockdown lines were accessed from the NCBI SRA at PRJNA197267 and mapped with STAR[26] v2.7.3a to the TE consensus sequence and r6.22 combined reference. DESeq2[92] was used to compare expression for each knockdown line to controls. TEs were considered to be regulated by the piRNA pathway in ovary if they were significantly upregulated after knockdown in at least 1 knockdown line (Wald test, two-sided adjusted $p < 0.05$).

**Reporting summary**. Further information on research design is available in the Nature Research Reporting Summary linked to this article.

## Data availability
The w1118 whole-genome sequencing data and w1118 adult testes bulk total RNA-seq data generated in this study have been deposited in the National Center for Biotechnology Information Sequence Read Archive (NCBI SRA) under accession code PRJNA727858. Previously published sequencing data used in this study is available at the NCBI SRA via project accessions PRJNA475132, PRJNA548742, PRJNA518743, and PRJNA197267. Source data are provided with this paper and via zenodo [https://doi.org/10.5281/zenodo.5554937][93].

## Code availability
All code[94] is provided as snakemake[95] workflows at github.com/Ellison-Lab/TestisTEs2021 and via zenodo [https://zenodo.org/record/5565461#.YWYGA3VKjeQ].

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

## Acknowledgements

The authors acknowledge the laboratory of Dr. Maureen Barr for use of their confocal microscope and Dr. Juan Wang for microscopy training and assistance. The authors also acknowledge the Office of Advanced Research Computing (OARC) at Rutgers, The State University of New Jersey for providing access to the Amarel cluster and associated research computing resources that have contributed to the results reported here. This research was supported by a National Institute of General Medical Sciences R01 award (GM140163) to C.E.E.

## Author contributions

M.A.L. and C.E.E. designed the study, interpreted the data, and wrote the paper. M.A.L. performed the microscopy work and computational analysis. W.C. prepared all Illumina sequencing libraries. C.E.E. acquired funding.

## Competing interests

The authors declare no competing interests.
