## [Peer Review File · Nature Communications]

A transposon expression burst accompanies the activation of Y chromosome fertility genes during *Drosophila* spermatogenesisREVIEWER COMMENTS

Reviewer #1 (Remarks to the Author):

The manuscript describes the analysis of transposable element (TE) expression in *Drosophila* testes from single-cell RNA-seq publicly available data coming from another study. This is a very interesting study.

The main results are that one cell-type (primary spermatocytes) specifically express some transposable elements. This suggests that in testes, TEs are allowed to be expressed during a precise window during which some genes associated to sperm maturation, flagella mobility and Y chromosome fertility factors are expressed. TEs expressed at this time contain more frequently at least one insertion on the Y, which seems in accordance with the apparent higher number of copies in males, and to the fact that there are less polymorphic insertions on the X (which spends less time in the male). Finally, genes involved in piRNA repression of TEs are downregulated during this developmental window, which can explain TE activation. In the discussion, the authors emphasize on a potential trade-off in the piRNA system that is mainly devoted to silencing some cryptic meiotic drive system in *D. melanogaster*.

Even if it is already known that TEs are generally more expressed in the male germline, the use of single cell RNAseq bring much information about the precise window and cell-types in which TEs expression occurs. How this can be related to other processes (piRNA regulation, de-repression of Y chromosome, involvement of the flamenco locus) is however still entangled.

The manuscript is well written. The study is very nice, and seriously done. Yet I am not competent to evaluate the quality of the method used for the ICA analysis.

I regret that sometimes not enough details are given in the specific analysis of the TEP-TE, including the method. How many families? Are they the ones that are presently active? Do they correspond to the ones that are most silenced in the ovaries? The results shown are quite synthetic and are given in percentage or ratio (Figure 5), which make them a bit fuzzy. The reasons why the TEP-TEs are activated remain elusive since this group is enriched in Y-linked insertions, or in flamenco-regulated TEs, but these do not concern all these TEs. Also, the piRNA pathway is downregulated at this time.

Paragraph 262-270: Is there any other hypothesis, beside the fact that TE would be more active in males, that could explain the reduced frequency of X-linked insertion?

The enhanced expression of some TEs in testes has also been observed in *D. simulans*, associated with reduced piRNA production (see Saint-Léandre. et al. doi:10.1093/gbe/evaa094 for example). How this can fit with the trade-off hypothesis if the loci mainly targeted by pi-cluster in *D. melanogaster* are absent in closely-related species?

-Line 329-330: “Why do spermatocytes show a weakened piRNA response at a developmental timepoint when the TE-rich Y chromosome is de-repressed?” It appears obvious to me that the weakening of the piRNA response allows the TEs to be more expressed, even if other processes are involved. Interestingly, a spatiotemporal window during which TEs are less repressed also exist in females. (DOI: 10.1093/nar/gky695)

There are some problems in figure order and naming, and in the gene names reported in dotplots.

Minor points are listed below.

-The authors complete the analysis of scRNA-seq with RNA-seq and WGS from the strain w1118. Yet it is not clear whether it is the same strain that have been used for the scRNA-seq.

- Figure 1B: The gene names are not all concordant with what is described in the text.

-Line 108: “The final four clusters (3,4,5,6) represent the majority of the filtered cells.” From the figure, clusters 3 and 4 are rather represented by few cells, compared to some other clusters.

-Lines120- 122: It seems from Figure 2 that the TEs specifically expressed in terminal epithelial cells are also expressed specifically in pigment cells.

-Supplementary Figure 1: Maybe the panels order may be changed to better fit the text (panels C and D cited before panels A and B). I did not detect any reference to supplementary figure 1 panels E to J.

-Line 166: Supplementary Figure 3A (not 2A) (and also line 170 for Sup. Figure 3B). From this figure, I do not have the feeling that the scores are highly reproducible as stated on the text.

-Line 186: it is not defined what a TEP is. (transposon expression program?)

-Figure 4: Legends for panels B and C are inverted.

-It seems that Supplementary Figure 4 is not referenced in the text.

-Supplementary Figure 8A is not really convincing. Again, some genes cited in the text do not appear in the graph

Reviewer #2 (Remarks to the Author):

The manuscript, “A burst of transposon expression accompanies the activation of Y chromosome fertility genes during *Drosophila* spermatogenesis” is a thorough study on the important topic of transposable element mobilization, one of the major sources of genome change at macro and micro levels. As stated by the authors there has been enormous progress on understanding TE mobilization and host control mechanisms in the *Drosophila* ovary, but very little is known about the male germline. The authors have used a variety of published works and their own extensive analysis and work in the study. The work that went into this should not be underestimated. Reanalysis of data, especially scRNA-Seq data is complicated by the relative youth of the field and the lack of commonly accepted data standards and tools. Reusing Mahadevaraju et al. and Witt et al. data from the reads is the appropriate approach and the concordance testing provides important validation of the results from those studies, in addition to the important specific focus on TEs. The highly expressed TEs found in the early spermatocytes are also found enriched on the Y chromosome and depleted on the X relative to other active TEs. Importantly, the TEs expression coincides with the activation of Y linked genes suggesting the exploitation of Y chromosome activation for TE activity. The sex chromosome aspect of the work is particularly interesting. Thorough work! The conclusions follow nicely from the data and which are presented in an intellectually stimulating, but dense, manner.

Major:

1) The manuscript is too dense for the general reader. There is a lot of technical information in the results section and the acronym use in the main text is excessive. The authors will reach more people by

toning down the technical details in the results by augmenting the methods. There are a few specific instances in the minor comments, but really the authors should ruthlessly purge jargon throughout.

2) If I understand correctly, the implication is the X chromosome insertions are reduced due to the monosomy for the X in the male germline. Is this a 50% reduction then? Couldn't X chromosome inactivation contribute to this effect as well? Any deficit on the 4th? This point is worth discussing in the manuscript, not just in the response to review. The sex chromosome (including the 4th) aspect of the paper is quite interesting and should be complete.

3) In the supplement it would be better to include all the differentially expressed genes in some sort of a spreadsheet, rather than a limited list. Public databases are not well-developed for scRNA, so these large tables are really needed until the time that EBI expands their efforts, or NCBI launches an effort. Data availability requires putting more in the supplement than is usually needed. I am certain that people will refer to these often and that including this information will result in more citations (still a good metric of utility).

4) I have some specific comments about the cluster calls. Before launching into some specific examples, I recognize that there is unavoidable ambiguity in scRNA cell cluster calls for a wide variety of reasons, including resolution decisions in the data handling. The goal here to convey that uncertainty, not to say that there is anything "wrong" with the calls the authors made. Some of this expanded text on clusters could detract from the main text and might be better in the methods or supplement.

The observation that the cluster 2 may represent spermatogonia that have just begun the transition to meiotic prophase or the very early spermatocyte is an important observation. The mean normalized UMI counts for cluster 2 correlating with earlier studies and E1 early spermatocyte of that study may not be a good criteria for calling them as 'spermatogonia' because, the cells selected for the current study (8,000 cells) are only a subset of the previous data. In the figures, the cluster 2 is specifically called as spermatogonia so the ambiguity in the text (Line 107) needs to be clarified. Fig.1.B, 2/spermatogonia has high aly and can expression relatives to 3/spermatocytes. Aly and can are definitive for spermatocytes. It would be useful to show some other genes for spermatogonia biomarkers expressed in different cell types' to provide an opportunity for the testis expert reader to distinguish these two clusters (another reason to include a full table of DEGs in clusters in the supplement).

The hub is extremely difficult to call unambiguously due to the limited number of cells and overlapping expression patterns with terminal epithelial cells and early cyst cells. In the text related to Fig.1 (Line 98), cluster 10 is defined as 'hub and terminal epithelial cells' because of high Fas3 expression. In order to be definitive about the 'hub' cell, it is necessary to sub cluster the cluster 10 and specifically confirm the identity of the hub with several (presence and absence) of marker genes. There is a recent preprint from the Fly Cell Atlas that might help, although it is understandable if the authors do not want to use prepublication data. Another option is to not mention hub at all. Similarly, cluster 1 is labeled as spermatogonia but it is explained as 'Cluster 1 contains germline stem cell and early spermatogonia' (Line 100). Unless confirmed, the 'stem cell' identity cannot be assigned. Stem cells and spermatogonia are not the same.

minor:

5) Specific nomenclature needs to be followed throughout the manuscript consistently. The experiments to draw important conclusions are based on several different data sets which is impressive but, it is very difficult to follow the results. One example, in Line 138, 'We additionally queried poly-A RNA-seq reads from w1118 testis to test if detected TE expression..'. In this case 'poly-A RNA-seq data' can be either single cell or bulk as both the methods use poly-A RNA so make a clear distinction. Similarly, when 'w1118' is mentioned, this could point to previous scRNA-Seq or bulk RNA-Seq from larval testis or new adult data using w1118 presented in the manuscript. Briefly, specify bulk or sc, larval or adult.

6) The cluster 3/spermatocytes have the most TE-derived UMIs per cell which is an important observation reported. The TE expression is high in cluster 3 but 'uniformly' may be the representative word as there is a huge set of low expression in lot of cells (in the middle Fig.2).

7) There is a lot of densely presented material in Lines 129-140 and supplementary Fig.2 regarding if TE expression is a consequence of chimeric transcripts produced by TE insertions within host genes. Even though the conclusions are well written, the context and the process of identifying the 'genic fusion' and 'genetic break points' are not. I had to read this several times.

8) The information in the legend for Supplementary Fig.2 is not enough to understand the figure. For example, in the distribution of sense-strand poly-A RNA-seq signal for single-isoform host gene mRNAs and detected, TEs are 'in bins' but the bins are not well defined.

9) I'm not sure what a TE-enriched gene expression program is. Is it the host gene expression program that is co-expressed with TEs? Explain the criteria for genes falling into 'TE-enriched gene expression program' for identifying the GEPs either in the main text or methods and not just the code.

10) The development of a pipeline for TE-GEP detection. This whole section (144 to line 177) can be moved to methods, as it is a distraction from the main focus of the paper. Maybe under the 'pipeline' development to identify GEP using Independent Component Analysis (ICA)?

11) I don't think people are used to thinking about TEs and either female or male biased. A little clarification would help. In many places "n-biased TEs" is shorthand for TEs with n-biased expression. For example, "Several other TEs in this GEP have previously been shown to be male-biased". Some important information could be expanded to bring along more readers. For example, "Interestingly, we find that these TEs are enriched for elements located within the flamenco piRNA cluster, which is involved in TE suppression in ovarian follicle cells", is about ovary in a paper focusing on testis. Use a few more words for interesting information when it is raised.

12) TEP is a little confusion because not all TE families with intact Y-linked insertions are members of the TEP.

13) Why is EACm used as a marker for TEP-expressing spermatocytes? Maybe a reference missing here.

14) The Line 226 has a subheading 'TEP-TEs are enriched on the Y chromosome' but the section has also contained evidence for the downregulation of piRNA pathway genes that coincides with the activation of Y, so it's a bit broader.

15) This is a great paper, make the conclusion of the study reflect that. The end is a bit abrupt.

Clerical:

16) Mahadevaraju et al. is listed as 2020 or 2021 depending on where it is found. And is listed as a preprint. It's out here <https://www.nature.com/articles/s41467-021-20897-y>

17) Fig.1 write out UMAP and other technical abbreviations once.

18) Fig.2. The Y-axis represents 'all the TEs', but it might be better to have the specific number of different families.

19) Suppl Fig.2. A and B. The X-axis is not completely clear, maybe "TE-derived UMI" for specificity.

20) Fig.3 There is possible confusion between general GEP and specific TE-GEP. GEP-27 contains almost four-fold more TEs than the next most TE-rich GEP. Consider subdividing

21) Fig.4.write out TEP once.

We would like to thank the reviewers for their time and valuable feedback. Please find our point-by-point responses below.

Reviewer #1

The manuscript describes the analysis of transposable element (TE) expression in *Drosophila testes* from single-cell RNA-seq publicly available data coming from another study. This is a very interesting study.

The main results are that one cell-type (primary spermatocytes) specifically express some transposable elements. This suggests that in testes, TEs are allowed to be expressed during a precise window during which some genes associated to sperm maturation, flagella mobility and Y chromosome fertility factors are expressed. TEs expressed at this time contain more frequently at least one insertion on the Y, which seems in accordance with the apparent higher number of copies in males, and to the fact that there are less polymorphic insertions on the X (which spends less time in the male). Finally, genes involved in piRNA repression of TEs are downregulated during this developmental window, which can explain TE activation. In the discussion, the authors emphasize on a potential trade-off in the piRNA system that is mainly devoted to silencing some cryptic meiotic drive system in *D. melanogaster*.

Even if it is already known that TEs are generally more expressed in the male germline, the use of single cell RNAseq bring much information about the precise window and cell-types in which TEs expression occurs. How this can be related to other processes (piRNA regulation, de-repression of Y chromosome, involvement of the flamenco locus) is however still entangled.

The manuscript is well written. The study is very nice, and seriously done. Yet I am not competent to evaluate the quality of the method used for the ICA analysis.

Major Points

1) I regret that sometimes not enough details are given in the specific analysis of the TEP-TE, including the method. How many families? Are they the ones that are presently active? Do they correspond to the ones that are most silenced in the ovaries? The results shown are quite synthetic and are given in percentage or ratio (Figure 5), which make them a bit fuzzy. The reasons why the TEP-TEs are activated remain elusive since this group is enriched in Y-linked insertions, or in flamenco-regulated TEs, but these do not concern all these TEs. Also, the piRNA pathway is downregulated at this time.

There are 75 TE families in the TEP (which we now call *module 27*, see response #12 below). 73 of the 75 TE families have polymorphic insertions in *D. melanogaster* according to the TIDAL-FLY database, suggesting they have recently been, or are currently active. We have

added this information to our revised manuscript in the results section titled “A TE-rich gene module is expressed in spermatocytes”. We also provide two Supplementary Data files with a complete list of the TE families and host genes from *module 27*.

We examined bulk RNA-seq data from RNAi knockdowns of various piRNA pathway components in the adult ovary and identified TEs that show significant upregulation upon disruption of the piRNA pathway as being ovary-silenced. These ovary silenced TEs are significantly over-represented among the 75 *module 27* TEs. We have added this information to our revised results section:

*Interestingly, the module 27 TEs are enriched for elements located within the flamenco piRNA cluster, which is involved in TE suppression in ovarian follicle cells (Brennecke et al. 2007) (Fisher’s Exact Test, two-sided $P=0.03$)(Supplementary Figure 9A). One such TE is gypsy, which is both silenced by flamenco and is also a member of module 27. Gypsy has been previously reported to be enriched on the Y chromosome of several *D. melanogaster* strains, which Chalvet et al proposed was because Y-linked copies of this TE were able to escape silencing by the ovary-dominant flamenco locus (Chalvet et al. 1998). Consistent with such a strategy, module 27 is also enriched for TEs that are strongly silenced by the ovarian piRNA pathway in general (Fisher’s Exact Test, two-sided $P=7.6e-4$) (Supplementary Figure 9B), raising the possibility that the expression of these TEs in the testes allows them to continue to mobilize in *D. melanogaster* despite being silenced in the ovary.*

We now also provide raw counts in addition to percentages and ratios when we mention the *module 27* TE families. We have also determined that 80% of all *module 27* TEs either have Y-linked copies or are flamenco-regulated. The most complete assembly of the Y chromosome that is currently available only covers ~36% of the total estimated length of this chromosome. Therefore, it is likely that other *module 27* TEs also have Y-linked copies whose locations are in unassembled portions of the chromosome. We mention this in our revised manuscript:

Altogether, 80% of the TEs in module 27 have at least one full-length copy within the assembled portion of the Y chromosome and/or are flamenco-regulated. This is likely an underestimate given estimates that ~60% of the Y chromosome remains unassembled.

2) Paragraph 262-270: Is there any other hypothesis, beside the fact that TE would be more active in males, that could explain the reduced frequency of X-linked insertion?

Reviewer #2 has proposed the interesting idea that Meiotic Sex Chromosome Inactivation (MSCI) could also explain the reduced frequency of X-linked insertions. We have updated the manuscript to address this possibility:

It is also possible that Meiotic Sex Chromosome Inactivation (MSCI), where the X chromosome is transcriptionally downregulated in primary spermatocytes (Mahadevaraju et al. 2021), could further impede the ability of TEs to insert on the X chromosome during spermatogenesis. To assess this possibility, we compared frequencies of polymorphic TE insertions on the 4th chromosome to those on the autosomes. The 4th chromosome is the ancestral X chromosome in Dipterans and has also been shown to undergo transcriptional downregulation in D. melanogaster spermatocytes, along with the X chromosome (Mahadevaraju et al. 2021). However, in contrast to the X chromosome, we found that the module 27 TEs do not show a reduction in polymorphic insertions on the 4th chromosome (Supplementary Figure 11). These results suggest that the reduction in X-linked insertions for the module 27 TEs is more likely to be due to the monosomy of the X, rather than X chromosome inactivation, however, either mechanism is consistent with our conclusion that the module 27 TEs show male-biased activity.

3) The enhanced expression of some TEs in testes has also been observed in D. simulans, associated with reduced piRNA production (see Saint-Léandre. et al. doi:10.1093/gbe/evaa094 for example). How this can fit with the trade-off hypothesis if the loci mainly targeted by pi-cluster in D. melanogaster are absent in closely-related species?

This is a good point which we now address in the discussion:

Like melanogaster, D. simulans testes also show a weakening of piRNA-mediated TE silencing. D. simulans testes have been shown to have many fewer TE-derived piRNAs and higher TE expression compared to ovaries (Saint-Leandre et al. 2020). However, D. simulans lacks both the su(Ste) and AT-chX satellites. Thus, either the tradeoff between satellite- and TE-derived piRNAs is unique to D. melanogaster or a different tradeoff is present in D. simulans, such as between piRNA production versus other small RNAs. For example, siRNAs generated by hairpin RNAs have recently been shown to play an important role in resolving intragenomic conflict in the D. simulans male germline (Lin et al. 2018). Alternatively, there may be no tradeoff at all and the downregulation of the piRNA pathway during spermatogenesis may serve some other, as yet undetermined, developmental function.

4) Line 329-330: “Why do spermatocytes show a weakened piRNA response at a developmental timepoint when the TE-rich Y chromosome is de-repressed? “ It appears obvious to me that the weakening of the piRNA response allows the TEs to be more expressed, even if other processes are involved. Interestingly, a spatiotemporal window during which TEs are less repressed also exist in females. (DOI: 10.1093/nar/gky695)

We agree that the weakening of the piRNA response is probably responsible for TE upregulation but it is strange that the piRNA response would be weakened at the same time that the Y chromosome fertility genes become activated. *A priori*, we would expect the opposite to happen: an upregulation of piRNA components in spermatocytes to aggressively silence Y-linked TEs. We have now added additional text to clarify our argument (see below) and, in the Introduction, we have cited the paper the reviewer mentions.

The downregulation of the piRNA pathway in spermatocytes is surprising. One would expect the piRNA pathway to be upregulated in these cells to safeguard against TE insertions resulting from de-repression of Y-linked TEs.

5) There are some problems in figure order and naming, and in the gene names reported in dotplots.

We have included *spn-E* and *Sox100B* in the dotplot in Figure 1B and removed *rdo*, which was not described in the text. We also now include *bol* in Supplementary Figure 8A. We have also revised all supplemental figures so that the order of the panels matches the order they are cited in the text.

Minor points are listed below.

6) The authors complete the analysis of scRNA-seq with RNA-seq and WGS from the strain w1118. Yet it is not clear whether it is the same strain that have been used for the scRNA-seq.

Yes, it is the same strain. We have clarified this in the text:

We reanalyzed 10x Genomics 3' single-cell expression data from a recent study examining sex chromosome gene expression in larval testes from D. melanogaster strain w1118 (Mahadevaraju et al. 2021).

7) Figure 1B: The gene names are not all concordant with what is described in the text.

We have added *spn-E* and *Sox100B* and removed *rdo*, which was not described in the text.

8) Line 108: "The final four clusters (3,4,5,6) represent the majority of the filtered cells." From the figure, clusters 3 and 4 are rather represented by few cells, compared to some other clusters.

We have rephrased this sentence to remove ambiguity regarding sizes of individual clusters.

The final four clusters (3, 4, 5, and 6) represent the remaining filtered cells ...

9) Lines 120- 122: It seems from Figure 2 that the TEs specifically expressed in terminal epithelial cells are also expressed specifically in pigment cells.

This is a good point. We have amended the description of TE expression across the dataset to be more specific and clear. We have also included a barplot as Figure 2B that compares the number of TE families expressed in at least half the cells of each cluster.

Most striking are the cells from cluster 3 spermatocytes, where 28 TE families show expression in more than half of the cells in the cluster (Figure 2B). In comparison, only two TE families are expressed by at least half of cluster 1 spermatogonial cells and only one TE family shows expression in at least half of each cyst cell cluster (Figure 2B). Four and five TE families are expressed by at least half of Terminal Epithelial cells or Pigment cells respectively, of which 3 families overlap (Figure 2B). Interestingly, the cluster 2 transitional spermatocytes have the next largest number of expressed TE families (18 families expressed by more than half of cells, Figure 2B) and there is a high degree of overlap between the families expressed in cluster 2 and cluster 3 (14 TE families expressed in more than half of cells in both clusters), consistent with the transcriptional activation of these TEs families coinciding with the developmental transition from spermatogonia to spermatocytes.

10) Supplementary Figure 1: Maybe the panels order may be changed to better fit the text (panels C and D cited before panels A and B). I did not detect any reference to supplementary figure 1 panels E to J.

We have revised all supplemental figures so that the order of the panels matches the order they are cited in the text and we have confirmed that all panels are cited at least once in the text of the manuscript.

11) Line 166: Supplementary Figure 3A (not 2A) (and also line 170 for Sup. Figure 3B). From this figure, I do not have the feeling that the scores are highly reproducible as stated on the text.

We appreciate this feedback and have included a more specific and quantitative description of the extent of the reproducibility of the method. To assess reproducibility, we compared module membership scores for each module across 3 replicate runs of the consensus ICA algorithm. The membership scores reflect the strength of the association of a gene with a module. We show that all of the modules used in our main text have membership scores that are significantly positively correlated with modules found in all 3 ICA replicate runs.

To assess the reproducibility of the cICA module detection approach, for all gene module membership scores among the modules used in the main results of the study we

calculated pairwise Spearman's correlation with all modules from the three replicate runs used to find parameters k and q (see 'Module Detection Parameters' below). We find that all main result modules were significantly positively correlated (Spearman's ρ p value < 0.05) with a module in all grid search replicate runs. (Supplementary Figure 3A).

12) Line 186: it is not defined what a TEP is. (transposon expression program?)

We have changed our terminology for detected co-expression modules based on the recommendation for reviewer #2 to reduce usage of unnecessary acronyms. Rather than “gene expression programs”, we now use “modules” throughout the text and we refer to the TE-rich module (previously described as the TEP) as *module 27*. We have also included additional text in the Results to describe the modules:

To gain additional insight into the biological context involving the upregulation of TEs in cluster 3 spermatocytes, we used the single-cell expression profiles of both host genes and TEs to infer co-expression modules. Co-expression modules are groups of genes and/or TEs with correlated expression patterns. Members of the same expression module are frequently co-regulated and member genes often have related functions. Clustering-based algorithms are commonly used for the identification of modules, however clustering approaches usually examine co-expression across all samples, which is not ideal for single-cell expression data, where co-expression patterns may be limited to specific cell types. For this reason, we decided to infer gene and TE co-expression modules using an approach that can identify local co-expression signatures existing in only a subset of cells. Independent Component Analysis (ICA) has previously been shown to have this property and, in general, it performs favorably compared to other module detection methods (Saelens, Cannoodt, and Saeys 2018).

13) Figure 4: Legends for panels B and C are inverted.

We have changed the order of the panels to reflect the legend and order in the text.

14) It seems that Supplementary Figure 4 is not referenced in the text.

We now cite this figure in the Methods section. After reordering the figures to reflect the order they are cited in, it is now Supplementary Figure 13.

15) Supplementary Figure 8A is not really convincing. Again, some genes cited in the text do not appear in the graph

The main conclusion we draw from Figure S8A is that several well-known regulators of spermatocyte transcriptional programs are highly expressed in cluster 3 spermatocytes. The

point of confusion may have been related to the fact that most of these genes also show high expression in cluster 2, which we previously referred to as spermatogonia. Based on comments from Reviewer #2, we have reanalyzed this cluster and conclude that it most likely represents very early spermatocytes that have just transitioned from spermatogonia. Our conclusion now makes more sense in light of this change: known transcriptional regulators of early spermatocytes show high expression in clusters #2 and #3, which represent early spermatocyte cells.

Furthermore, we have confirmed that *can*, *sa*, *bol*, *aly*, *wuc*, *kmg*, *kl-2*, *kl-3*, *kl-*, *tplus3a*, and *tplus3b* are significantly differentially expressed in cluster 3 spermatocytes compared to all other cells and added these details to the text to make this explicit:

Two testis-specific TBP associated factors (TAFs), can and sa, are members of module 27 and are differentially expressed (T-test, two-sided adjusted $P < 0.05$) in cluster 3. Testis-specific Meiotic Arrest Complex (tMAC) components aly and wuc, which promote transcription of spermatocyte-specific genes by activating alternative promoters (Lu et al. 2020), are members of module 27, as well as kmg, which blocks promiscuous activation of genes by tMAC (Kim et al. 2017). Each of these is also significantly differentially expressed in cluster 3 (T-test, two-sided adjusted $P < 0.05$). This supports our analysis suggesting cluster 3 is predominantly composed of primary spermatocytes.

...

The 6 Y-linked genes, bol, tplus3a, and tplus3b, are differentially expressed (T-test, two-sided adjusted $P < 0.05$) in cluster 3 spermatocytes (Figure 4B, Supplementary Figure 9A).

We have also adjusted the size scale of the dotplot points to more easily visually identify expression of spermatocyte specific genes in cluster 3. Additionally, we have added *bol*, which was referenced in the text but missing from the plot.

Reviewer #2

The manuscript, "A burst of transposon expression accompanies the activation of Y chromosome fertility genes during *Drosophila* spermatogenesis" is a thorough study on the important topic of transposable element mobilization, one of the major sources of genome change at macro and micro levels. As stated by the authors there has been enormous progress on understanding TE mobilization and host control mechanisms in the *Drosophila* ovary, but very little is known about the male germline. The authors have used a variety of published works and their own extensive analysis and work in the study. The work that went into this should not be underestimated. Reanalysis of data, especially scRNA-Seq data is complicated by the relative youth of the field and the lack of commonly accepted data standards and tools. Reusing Mahadevaraju et al. and Witt et al. data from the reads is the appropriate approach and the concordance testing provides important validation of the results from those studies, in addition to the important specific focus on TEs. The highly expressed TEs found in the early spermatocytes are also found enriched on the Y chromosome and depleted on the X relative to other active TEs. Importantly, the TEs expression coincides with the activation of Y linked genes suggesting the exploitation of Y chromosome activation for TE activity. The sex chromosome aspect of the work is particularly interesting. Thorough work! The conclusions follow nicely from the data and which are presented in an intellectually stimulating, but dense, manner.

Major:

1) The manuscript is too dense for the general reader. There is a lot of technical information in the results section and the acronym use in the main text is excessive. The authors will reach more people by toning down the technical details in the results by augmenting the methods. There are a few specific instances in the minor comments, but really the authors should ruthlessly purge jargon throughout.

The reviewer raises a good point here. We have extensively revised the Results section, including moving several full paragraphs to the Methods. We also made the following changes across the whole manuscript, which we hope will help to make it easier to read and more accessible: We no longer use the acronyms "GEP" and "TEP". Instead, we now refer to "gene expression programs" as *co-expression modules*, or simply as *modules*. Additionally, what we have previously termed the "TEP", we now refer to as *module 27*.

2) If I understand correctly, the implication is the X chromosome insertions are reduced due to the monosomy for the X in the male germline. Is this a 50% reduction then? Couldn't X chromosome inactivation contribute to this effect as well? Any deficit on the 4th? This point is worth discussing in the manuscript, not just in the response to review.

The sex chromosome (including the 4th) aspect of the paper is quite interesting and should be complete.

That is correct -- the X represents a smaller target for TE insertions in the testes compared to the ovaries. We observe a ~23% reduction in X-linked insertions for the *module 27* TEs rather than a 50% reduction. However, a 50% reduction would only occur if the *module 27* TEs were completely inactive in females. The fact that we do not see this large of a reduction suggests that these TEs are also occasionally active in females as well.

We have repeated our analysis for the 4th chromosome, however, we do not see a reduction in insertions on this chromosome for the *module 27* TEs (see new Supplementary Figure 11). We therefore favor the monosomy explanation but have revised the manuscript to acknowledge both possibilities:

*It is also possible that Meiotic Sex Chromosome Inactivation (MSCI), where the X chromosome is transcriptionally downregulated in primary spermatocytes (Mahadevaraju et al. 2021), could further impede the ability of TEs to insert on the X chromosome during spermatogenesis. To assess this possibility, we compared frequencies of polymorphic TE insertions on the 4th chromosome to those on the autosomes. The 4th chromosome is the ancestral X chromosome in Dipterans and has also been shown to undergo transcriptional downregulation in *D. melanogaster* spermatocytes, along with the X chromosome (Mahadevaraju et al. 2021). However, in contrast to the X chromosome, we found that the module 27 TEs do not show a reduction in polymorphic insertions on the 4th chromosome (Supplementary Figure 11). These results suggest that the reduction in X-linked insertions for the module 27 TEs is more likely to be due to the monosomy of the X, rather than X chromosome inactivation, however, either mechanism is consistent with our conclusion that the module 27 TEs show male-biased activity.*

3) In the supplement it would be better to include all the differentially expressed genes in some sort of a spreadsheet, rather than a limited list. Public databases are not well-developed for scRNA, so these large tables are really needed until the time that EBI expands their efforts, or NCBI launches an effort. Data availability requires putting more in the supplement than is usually needed. I am certain that people will refer to these often and that including this information will result in more citations (still a good metric of utility).

This is a good suggestion. We have added supplementary data file #3, which is a tab delimited file containing differentially expressed genes for each cluster.

4) I have some specific comments about the cluster calls. Before launching into some specific examples, I recognize that there is unavoidable ambiguity in scRNA cell cluster

calls for a wide variety of reasons, including resolution decisions in the data handling. The goal here to convey that uncertainty, not to say that there is anything “wrong” with the calls the authors made. Some of this expanded text on clusters could detract from the main text and might be better in the methods or supplement.

The observation that the cluster 2 may represent spermatogonia that have just begun the transition to meiotic prophase or the very early spermatocyte is an important observation. The mean normalized UMI counts for cluster 2 correlating with earlier studies and E1 early spermatocyte of that study may not be a good criteria for calling them as ‘spermatogonia’ because, the cells selected for the current study (8,000 cells) are only a subset of the previous data. In the figures, the cluster 2 is specifically called as spermatogonia so the ambiguity in the text (Line 107) needs to be clarified. Fig.1.B, 2/spermatogonia has high aly and can expression relatives to 3/spermatocytes. Aly and can are definitive for spermatocytes. It would be useful to show some other genes for spermatogonia biomarkers expressed in different cell types’ to provide an opportunity for the testis expert reader to distinguish these two clusters (another reason to include a full table of DEGs in clusters in the supplement).

The hub is extremely difficult to call unambiguously due to the limited number of cells and overlapping expression patterns with terminal epithelial cells and early cyst cells. In the text related to Fig.1 (Line 98), cluster 10 is defined as ‘hub and terminal epithelial cells’ because of high Fas3 expression. In order to be definitive about the ‘hub’ cell, it is necessary to sub cluster the cluster 10 and specifically confirm the identity of the hub with several (presence and absence) of marker genes. There is a recent preprint from the Fly Cell Atlas that might help, although it is understandable if the authors do not want to use prepublication data. Another option is to not mention hub at all. Similarly, cluster 1 is labeled as spermatogonia but it is explained as ‘Cluster 1 contains germline stem cell and early spermatogonia’ (Line 100). Unless confirmed, the ‘stem cell’ identity cannot be assigned. Stem cells and spermatogonia are not the same.

We appreciate the suggestions that will better convey the uncertainty inherent to cell identity calling. To this end we have removed references to germline stem cells in cluster 1, as this cluster is more strongly marked by spermatogonial markers. For similar reasons, we have removed references to hub cells in our description of cluster 10. We have included an additional figure (Supplementary Figure 1C) showing a dotplot of an expanded set of spermatogonial markers and supplied a tabular file with differential expression results as Supplementary Data 3. We have additionally included the following text:

Cluster 2 is most transcriptionally similar to the G spermatogonia cluster identified by Mahadevaraju et al. but mean normalized unique molecular identifier (UMI) counts for

this cluster also correlate well with that study's E1 early spermatocyte cluster (Supplementary Figure 1B). Furthermore, this cluster expresses spermatogonial markers such as bam as well as spermatocyte markers such as aly, which respectively are required for germline stem cell differentiation and initiation of a primary spermatocyte transcription program (Supplementary Figure 1C). This observation suggests that our cluster 2 may represent spermatogonia just beginning the transition to meiotic prophase or very early spermatocytes. We therefore refer to cluster 2 cells as transitional spermatocytes.

minor:

5) Specific nomenclature needs to be followed throughout the manuscript consistently. The experiments to draw important conclusions are based on several different data sets which is impressive but, it is very difficult to follow the results. One example, in Line 138, 'We additionally queried poly-A RNA-seq reads from w1118 testis to test if detected TE expression..'. In this case 'poly-A RNA-seq data' can be either single cell or bulk as both the methods use pol-A RNA so make a clear distinction. Similarly, when 'w1118' is mentioned, this could point to previous scRNA-Seq or bulk RNA-Seq from larval testis or new adult data using w1118 presented in the manuscript. Briefly, specify bulk or sc, larval or adult.

In every place that we mention RNA-seq data, we now also specify whether the data are bulk or single-cell and larval or adult.

6) The cluster 3/spermatocytes have the most TE-derived UMIs per cell which is an important observation reported. The TE expression is high in cluster 3 but 'uniformly' may be the representative word as there is a huge set of low expression in lot of cells (in the middle Fig.2).

We appreciate this point. We have changed the wording to more accurately reflect that not all cells in this cluster express TEs highly and that not all TE families in our set of consensus sequences are highly expressed. We also now quantify the number of TE families expressed in at least half the cells of each cluster (see Figure 2B) as a way to compare the number of expressed TEs between clusters while also accounting for the fact that TE expression may not be uniform across the cells within a given cluster:

Most striking are the cells from cluster 3 spermatocytes, where 28 TE families show expression in more than half of the cells in the cluster (Figure 2B). In comparison, only two TE families are expressed by at least half of cluster 1 spermatogonial cells and only one TE family shows expression in at least half of each cyst cell cluster (Figure 2B). Four

and five TE families are expressed by at least half of Terminal Epithelial cells or Pigment cells respectively, of which 3 families overlap (Figure 2B). Interestingly, the cluster 2 transitional spermatocytes have the next largest number of expressed TE families (18 families expressed by more than half of cells, Figure 2B) and there is a high degree of overlap between the families expressed in cluster 2 and cluster 3 (14 TE families expressed in more than half of cells in both clusters), consistent with the transcriptional activation of these TE families coinciding with the developmental transition from spermatogonia to spermatocytes. However, cluster 3 spermatocytes have the most TE-derived UMIs per cell, for both depth-normalized and raw UMI counts (Supplementary Figure 2A, 2B).

7) There is a lot of densely presented material in Lines 129-140 and supplementary Fig.2 regarding if TE expression is a consequence of chimeric transcripts produced by TE insertions within host genes. Even though the conclusions are well written, the context and the process of identifying the 'genic fusion' and 'genetic break points' are not. I had to read this several times.

We have added additional explanatory text to the Results section:

We next assessed whether TE fragments nested in other cellular RNAs may be artificially increasing measurements of TE expression in the testes. If a TE fragment were present within a highly expressed host gene (in the UTR, for example), the RNA-seq reads from the fragment would get mapped to the TE consensus sequence, thus artificially inflating the expression level for that TE. We used two approaches to determine whether this phenomenon was affecting our estimates of TE expression: (1) we explicitly searched for chimeric RNA-seq reads, where part of the read aligned to a TE and another part aligned to a host gene, and (2) we examined RNA-seq read coverage along the TE consensus sequence. The full-length TE consensus should show non-uniform sequencing coverage with the region corresponding to the TE fragment showing much higher coverage than the rest of the TE.

We have also revised the Figure S2 legend to clarify these points (see bold below):

Supplementary Figure 2. TE expression in scRNA-seq data from larval testes. A) Violin plots show distributions of raw TE-mapping UMI counts in each w1118 L3 testis cell cluster. TE counts vary significantly among the clusters (Kruskal-Wallis, $p < 2.2e-16$). B) Violin plots show distributions of depth normalized TE-mapping UMI counts in each w1118 L3 testis cell cluster. TE counts vary significantly among the clusters (Kruskal-Wallis, $p < 2.2e-16$). C) Scatterplots show pseudo-bulk expression derived from our w1118 scRNA pipeline (see Methods) versus bulk expression for four w1118 testis poly-A RNA-seq replicates generated by Mahadevaraju et al. 2021 (see Methods). Each

replicate shows strong correlation with pseudo-bulk (all Pearson's $R \geq 0.89$, $P < 2.2e-16$). D) Scatterplots show same analysis described in D but restricted to TEs (all Pearson's $R \geq 0.85$, $P < 2.2e-16$). **E) Heatmaps show distribution of sense-strand bulk poly-A RNA-seq signal for single-isoform host gene mRNAs (left) and detected TEs (right). Larval bulk poly-A RNA-seq data generated by Mahadevaraju et al. 2021 was mapped to each feature (see Methods) and divided into bins representing one tenth the full length of each feature.** F) Boxplots show standard deviations of expression across bins for host genes (gray) and TEs (red). Three of four replicates show no significant difference in variability of poly-A signal across bins within features (Wilcoxon rank-sum test $P > 0.05$). Replicate 3 shows a significant difference (Wilcoxon rank-sum test, two-sided $P = 0.047$). For all boxplots, midline represents median, box represents interquartile range (IQR), and whiskers extend > 1.5 IQR from the upper or lower quartile. **G) Bar plot shows number of TE-gene chimeric transcripts reproducibly found in bulk poly-A RNA-seq data, for all TEs with a least one chimeric transcript identified in at least two of the replicate RNA-seq datasets.** **H) For each TE introduced in G, the y-axis position of each point represents the number of uniquely-mapping chimeric reads detected by STAR that support each chimeric transcript (see Methods).**

We have clarified that we queried putative chimeric reads identified by STAR in the main text:

We additionally queried the chimeric reads identified by the STAR aligner from w1118 larval testis poly-A selected bulk RNA-seq dataset (the same strain used for the scRNA-seq data).

We have additionally identified in the methods the STAR output file from which we identified these reads:

We identified evidence of chimeric TE transcripts from the "Chimeric.out.junction" output file.

8) The information in the legend for Supplementary Fig.2 is not enough to understand the figure. For example, in the distribution of sense-strand poly-A RNA-seq signal for single-isoform host gene mRNAs and detected, TEs are 'in bins' but the bins are not well defined.

We have added additional information to the legend to clarify the definition of bins:

Heatmaps show distribution of sense-strand poly-A RNA-seq signal for single-isoform host gene mRNAs (left) and detected TEs (right). Larval poly-A RNA-seq data generated

by Mahadevaraju et al. 2021 was mapped to each feature (see Methods) and divided into bins representing one tenth the full length of each feature.

9) I'm not sure what a TE-enriched gene expression program is. Is it the host gene expression program that is co-expressed with TEs? Explain the criteria for genes falling into 'TE-enriched gene expression program' for identifying the GEPs either in the main text or methods and not just the code.

In order to simplify the manuscript (and reduce acronym usage), we now simply call these "co-expression modules". The TEs are treated the same as genes for the purpose of identifying these modules and a given module could potentially contain no TEs (i.e. host genes only), or only TEs (i.e. no host genes). After identifying the modules, we then quantified the number of TEs present within each one, which is how we identified the TE-rich module that is expressed in early spermatocytes. We have added additional details to the methods to explain this process.

10) The development of a pipeline for TE-GEP detection. This whole section (144 to line 177) can be moved to methods, as it is a distraction from the main focus of the paper. Maybe under the 'pipeline' development to identify GEP using Independent Component Analysis (ICA)?

We have move these lines to the Methods and replaced them with the text below:

To gain additional insight into the biological context involving the upregulation of TEs in cluster 3 spermatocytes, we used the single-cell expression profiles of both host genes and TEs to infer co-expression modules. Co-expression modules are groups of genes and/or TEs with correlated expression patterns. Members of the same expression module are frequently co-regulated and member genes often have related functions. Clustering-based algorithms are commonly used for the identification of modules, however clustering approaches usually examine co-expression across all samples, which is not ideal for single-cell expression data, where co-expression patterns may be limited to specific cell types. For this reason, we decided to infer gene and TE co-expression modules using an approach that can identify local co-expression signatures existing in only a subset of cells. Independent Component Analysis (ICA) has previously been shown to have this property and, in general, it performs favorably compared to other module detection methods (Saelens, Cannoodt, and Saeys 2018). We implemented a consensus ICA approach (Kotliar et al. 2019) and selected parameters that resulted in modules representing a wide range of biological processes while also showing minimal overlap in terms of their gene content (see Methods). The modules resulting from this approach were reproducibly identified across replicate runs of consensus ICA (Supplementary Figure 3A) and ranged in size from 10 to over 600 genes, with 72% (65 out of 90) of identified modules containing 200 or fewer genes

(Supplementary Figure 3B). 64% (58 out of 90) percent of identified modules were enriched at $p < 0.05$ for a Biological Process Gene Ontology (GO) term not enriched in any other module (Supplementary Figure 3C).

11) I don't think people are used to thinking about TEs and either female or male biased. A little clarification would help. In many places "n-biased TEs" is shorthand for TEs with n-biased expression. For example, "Several other TEs in this GEP have previously been shown to be male-biased". Some important information could be expanded to bring along more readers. For example, "Interestingly, we find that these TEs are enriched for elements located within the flamenco piRNA cluster, which is involved in TE suppression in ovarian follicle cells", is about ovary in a paper focusing on testis. Use a few more words for interesting information when it is raised.

We have now clarified our use of "male-biased" or "female-biased" by explicitly referring to either biased expression or biased insertion activity. We have also clarified our reference to *flamenco* by explaining a prior study that observed an enrichment of the *flamenco*-silenced *gypsy* TE on the Y chromosome of several *D. melanogaster* strains:

Interestingly, the module 27 TEs are enriched for elements located within the flamenco piRNA cluster, which is involved in TE suppression in ovarian follicle cells (Brennecke et al. 2007) (Fisher's Exact Test, two-sided $P=0.03$)(Supplementary Figure 9A). One such TE is gypsy, which is both silenced by flamenco and is also a member of module 27. Gypsy has been previously reported to be enriched on the Y chromosome of several D. melanogaster strains, which Chalvet et al proposed was because Y-linked copies of this TE were able to escape silencing by the ovary-dominant flamenco locus (Chalvet et al. 1998).

12) TEP is a little confusion because not all TE families with intact Y-linked insertions are members of the TEP.

We no longer use the term "TEP" and instead simply refer to the TE-rich module that is expressed in early spermatocytes as *module 27*.

We are also intrigued by the observation that not all TE families with Y-linked insertions are members of the module. It is possible that only a subset of TE insertion locations on the Y chromosome are permissive to TE transcription, possibly due to the presence of nearby regulatory elements or transcriptionally activating chromatin marks. As improved Y assemblies become available, determining the reasons for this effect will become an interesting avenue for study. We address this issue in the main text:

Notably, not all TE families with intact Y-linked insertions are members of module 27, suggesting that additional features beyond Y-linkage, such as specific regulatory elements, are required for TEs to exploit this opportunity.

13) Why is EAC_{hm} used as a marker for TEP-expressing spermatocytes? Maybe a reference missing here.

EAC_{hm} is a member of the TE-rich module 27 and a gene whose expression, based on the scRNA-seq data, is very high and also very specific for the cells where module 27 is expressed. We are using it as a marker gene for module 27-expressing cells. We have added the line below in order to clarify this point:

We identified the host gene EAC_{hm} as a marker gene that shows high specificity for module 27-expressing spermatocytes (Figure 4A).

14) The Line 226 has a subheading ‘TEP-TEs are enriched on the Y chromosome’ but the section has also contained evidence for the downregulation of piRNA pathway genes that coincides with the activation of Y, so it’s a bit broader.

We have updated the subheading to better reflect the content of the section.

Y chromosome activation and host defense downregulation coincide with module 27 TE expression

15) This is a great paper, make the conclusion of the study reflect that. The end is a bit abrupt.

We have expanded the final paragraph of the discussion:

*In summary, this work provides detailed insight into the expression dynamics of TEs in the male germline of *Drosophila*. We developed a novel approach for identifying gene/TE co-expression modules, which we used to identify a TE-rich expression module that is active in early spermatocytes. These results suggest that spermatocytes represent a permissive niche for TE expression due to the downregulation of piRNA pathway components and the transcriptional activation of the TE-rich Y chromosome in these cells. Additional work to improve the Y chromosome assembly and uncover the mechanisms underlying the transcriptional activation of the Y chromosome will be critical for understanding the burst of TE expression that we have identified in spermatocytes. Furthermore, future work investigating the peculiarities of TE silencing in the testes will help shed light upon the tradeoffs and constraints imposed by the various roles of*

piRNAs in the male germline, including host gene regulation, TE silencing, and the resolution of intragenomic conflicts.

Clerical:

16) Mahadevaraju et al. is listed as 2020 or 2021 depending on where it is found. And is listed as a preprint. It's out here <https://www.nature.com/articles/s41467-021-20897-y>

We have updated the citation for this paper to consistently refer to the peer-reviewed version.

17) Fig.1 write out UMAP and other technical abbreviations once.

We have written out technical abbreviations in the main text and captions.

18) Fig.2. The Y-axis represents 'all the TEs', but it might be better to have the specific number of different families.

We have amended the figure caption (see new text below) and Y axis label to reflect the number of specific families.

Heatmap shows scaled expression level of 128 transposable element families detected in this dataset across all cells.

19) Suppl Fig.2. A and B. The X-axis is not completely clear, maybe "TE-derived UMI" for specificity.

We have amended the axis labels for these figures to make the origin of the UMIs clear.

20) Fig.3 There is possible confusion between general GEP and specific TE-GEP. GEP-27 contains almost four-fold more TEs than the next most TE-rich GEP. Consider subdividing

Our module detection approach is agnostic as to whether a given module contains genes, TEs, or both genes and TEs. After inferring module membership for the combined set of genes and TEs, we then scanned through all modules to determine which of them contained TEs and then calculated the abundance of TEs within each TE-containing module. Module 27 stood out for precisely the reason mentioned by the reviewer: it contains many more TEs than any other module. We have revised the Methods section to clarify this aspect of our module detection approach:

We performed ICA factorization using expression values for highly variable genes identified by scanpy as well as the 128 TE families that passed initial filtering. Therefore, resulting modules may contain exclusively TEs or genes or a mix of both types of feature.

21) Fig.4.write out TEP once.

We have changed our terminology for detected co-expression modules. We now refer to these simply as 'modules' throughout the text and we refer to the TEP as *module 27*.

REVIEWERS' COMMENTS

Reviewer #1 (Remarks to the Author):

I found the revised version much easier to read. It is indeed a very interesting paper.

I thank the authors, who have been careful to answer to all the requests and to make deep changes when necessary. I have no other comment to add.

Just maybe in the figure 3, there is still some acronym "GEP" that could be changed to "module" to better fit the text and figure legend.

Reviewer #3 (Remarks to the Author):

looks ready to go

We have revised our manuscript in accordance with the comments from the reviewers below.

REVIEWERS' COMMENTS

Reviewer #1 (Remarks to the Author):

I found the revised version much easier to read. It is indeed a very interesting paper. I thank the authors, who have been careful to answer to all the requests and to make deep changes when necessary. I have no other comment to add.

Just maybe in the figure 3, there is still some acronym "GEP" that could be changed to "module" to better fit the text and figure legend.

We have revised Figure 3 to replace the GEP acronym.

Reviewer #3 (Remarks to the Author):

looks ready to go

Thank you.